# Boosting ATM activity alleviates aging and extends lifespan in a mouse model of progeria

Minxian Qian[1,2,3†], Zuojun Liu[1,2,3†], Linyuan Peng[1,2,3], Xiaolong Tang[1,2,3], Fanbiao Meng[1,2,3], Ying Ao[1,3], Mingyan Zhou[1,2,3], Ming Wang[1,2,4], Xinyue Cao[1,2,3], Baoming Qin[4], Zimei Wang[1,3], Zhongjun Zhou[5], Guangming Wang[6,7], Zhengliang Gao[7,8], Jun Xu[6], Baohua Liu[1,2,3]*

[1]Guangdong Key Laboratory of Genome Stability and Human Disease Prevention, Shenzhen University Health Science Center, Shenzhen, China; [2]Medical Research Center, Shenzhen University Health Science Center, Shenzhen, China; [3]Department of Biochemistry and Molecular Biology, Shenzhen University Health Science Center, Shenzhen, China; [4]South China Institute for Stem Cell Biology and Regenerative Medicine, Guangzhou Institutes of Biomedicine and Health, Chinese Academy of Sciences, Guangzhou, China; [5]School of Biomedical Sciences, Li Ka Shing Faculty of Medicine, University of Hong Kong, Hong Kong, China; [6]East Hospital, Tongji University School of Medicine, Shanghai, China; [7]Shanghai Tenth People's Hospital, Tongji University School of Medicine, Shanghai, China; [8]Advanced Institute of Translational Medicine, Tongji University, Shanghai, China

*For correspondence:
ppliew@szu.edu.cn

†These authors contributed equally to this work

Competing interests: The authors declare that no competing interests exist.

**Abstract** DNA damage accumulates with age (Lombard et al., 2005). However, whether and how robust DNA repair machinery promotes longevity is elusive. Here, we demonstrate that ATM-centered DNA damage response (DDR) progressively declines with senescence and age, while low dose of chloroquine (CQ) activates ATM, promotes DNA damage clearance, rescues age-related metabolic shift, and prolongs replicative lifespan. Molecularly, ATM phosphorylates SIRT6 deacetylase and thus prevents MDM2-mediated ubiquitination and proteasomal degradation. Extra copies of *Sirt6* extend lifespan in *Atm-/-* mice, with restored metabolic homeostasis. Moreover, the treatment with CQ remarkably extends lifespan of *Caenorhabditis elegans*, but not the *ATM-1* mutants. In a progeria mouse model with low DNA repair capacity, long-term administration of CQ ameliorates premature aging features and extends lifespan. Thus, our data highlights a pro-longevity role of ATM, for the first time establishing direct causal links between robust DNA repair machinery and longevity, and providing therapeutic strategy for progeria and age-related metabolic diseases.
DOI: https://doi.org/10.7554/eLife.34836.001

## Introduction

A variety of metabolic insults can generate DNA lesions in mammalian cells, which, if incorrectly repaired, can lead to somatic mutations and cell transformation (*Vijg, 2014*). If unrepaired, such lesions can accumulate and constantly activate the DNA damage response (DDR), a unique feature and mechanism of senescence (*Halliwell and Whiteman, 2004*; *Tanaka et al., 2006*). Ataxia telangiectasia mutated (ATM), a serine/threonine protein kinase, is one of the key regulators of DDR (*Guleria and Chandna, 2016*). Upon DNA damage, self-activated ATM phosphorylates downstream transducers and effectors, promoting DNA repair (*Bakkenist and Kastan, 2003*; *Paull, 2015*). H2AX

**eLife digest** As cells live and divide, their genetic material gets damaged. The DNA damage response is a network of proteins that monitor, recognize and fix the damage, which is also called DNA lesions. For example, an enzyme called ATM senses when DNA is broken and then begins a process that will get it repaired, while another enzyme known as SIRT6 participates in the actual mending process.

When organisms get older, the repair machinery becomes less efficient, and the number of DNA lesions and errors increases. The accumulation of DNA damage may cause the 'symptoms' of old age – from cancer, to wrinkles and the slowing down of the body's chemical processes. In fact, individuals with defective ATMs (who thus struggle to repair their DNA) age abnormally fast; conversely, SIRT6 promotes longevity. If declining repair mechanisms cause aging, would boosting the DNA damage response slow down this process?

Chloroquine is a drug used to combat malaria, but it can also enhance the activity of ATM without damaging DNA. Qian, Liu et al. show that chloroquine helps cells remove broken DNA and keep dividing for longer. In animals, the drug increases the lifespan of worms and prolongs the lives of mice who have mutations that make them age quicker.

Qian, Liu et al. also demonstrate that ATM works by chemically altering the pro-longevity enzyme SIRT6. These changes make SIRT6 more stable, and keep it safe from cellular processes that destroy it. In addition, mice that are genetically engineered to lack ATM can survive for longer if they also produce extra SIRT6. These experiments show that enhancing the DNA damage response can slow down aging, thus linking the DNA repair machinery to longevity.

Progeria is a group of rare genetic conditions with inefficient DNA repair; people with progeria age fast and die young. The results by Qian, Liu et al., if confirmed in humans, could provide a new way of treating these diseases.

DOI: https://doi.org/10.7554/eLife.34836.002

is a well-documented phosphorylation target of ATM; phosphorylated H2AX at S139 ($\gamma$H2AX) is widely recognized as a hallmark of DNA damage (*Burma et al., 2001*). Accompanied by decline in DNA repair function, $\gamma$H2AX-enriched DNA damage foci accumulate in senescent cells and in tissues from aged animals (*Maslov and Vijg, 2009*), supporting causal links between defective DDR and aging. In human fibroblasts, a dramatic decline of homologous recombination (HR) efficiency, attributable to defective recruitment of Rad51, has been observed (*Li et al., 2016*). Similar defects in HR also have been observed in Hutchinson-Gilford progeria syndrome (HGPS), which is predominantly caused by a *LMNA* C1024T mutation (*Liu et al., 2005*).

In addition to DNA damage accumulation, inherited loss-of-function mutations in essential components of DNA repair machinery accelerate aging in humans and mice (*Hoeijmakers, 2009b*). Patients suffering from ataxia telangiectasia (A-T) develop prominent aging features in their second decades (*Boder and Sedgwick, 1958*; *Shiloh and Lederman, 2017*). Werner syndrome, Bloom's syndrome and Rothmund-Thomson syndrome are all progeria syndromes caused by mutations of genes that directly regulate DNA repair (*Balajee et al., 1999*; *Cooper et al., 2000*; *Lebel et al., 1999*; *Li and Comai, 2000*). Homozygous disruption of *Atm* in mice causes many premature aging features of A-T, such as growth retardation, infertility, neurodegeneration, immunodeficiency and cancer predisposition (*Barlow et al., 1996*). Mouse models deficient in DNA repair factors, including DNA-PKcs, Ku70, Ku80, DNA ligase IV, Artemis or Ercc1 etc., phenocopy premature aging features (*Hasty, 2005*; *Hoeijmakers, 2009a*), supporting the suggestion that defects in DNA repair accelerate aging. However, whether and how robust DNA repair machinery promotes longevity is poorly understood.

Metabolic disturbance is another antagonistic hallmark of aging (*López-Otín et al., 2013*). Although DNA repair deficiency is implicated in aging and age-related diseases including metabolic disorders (*López-Otín et al., 2016*; *Shimizu et al., 2014*), the mechanistic link between decreased DNA repair machinery and metabolic reprogramming during aging is poorly understood. Notably, in response to oxidative stress, ATM phosphorylates Hsp27, shifting glucose metabolism from glycolysis to the pentose phosphate pathway (PPP) (*Cosentino et al., 2011*; *Krüger and Ralser, 2011*).

Inactivating ATM enhances glucose and glutamine consumption by inhibiting P53 and upregulating c-MYC (*Aird et al., 2015*). However, the role of ATM in age-onset metabolic disturbances is as yet unclear.

Here, we identified a progressive decline in ATM-centered DNA repair machinery during aging, along with shunted glucose metabolism to glycolysis. DNA damage-free activation of ATM by chloroquine (CQ) promotes DNA damage clearance, rescues age-related metabolic shift, and alleviates cellular senescence. Mechanistically, ATM phosphorylates and stabilizes pro-longevity protein SIRT6. Extra copies of *Sirt6* attenuate metabolic abnormality and extend lifespan in *Atm-/-* mice. Importantly, long-term treatment of CQ restores metabolic reprogramming and extends the lifespan of nematodes and a progeria mouse model.

## Results

### ATM activation alleviates replicative senescence

In searching for genes/pathways that drive senescence, we employed human primary endothelial cells, which underwent replicative senescence at passage 21, with increased p21 expression and β-galactosidase activity (*Figure 1—figure supplement 1a–b*). By RNAseq analysis, a gradual decline of ATM-centered DNA repair machinery was identified (*Figure 1—figure supplement 1c–e*). Western blotting analysis confirmed progressively downregulated protein levels of ATM and its downstream target NBS1 and RAP80 in senescent human skin fibroblasts (HSFs) (*Figure 1a*). Mouse embryonic fibroblasts (MEFs) with limited growth capacity and senescent phenotypes when cultured in vitro (*Parrinello et al., 2003*; *Samper et al., 2003*; *Sherr and DePinho, 2000*), and brain tissues from aged mice also showed progressive decline of ATM, NBS1, and RAP80 (*Figure 1b–c*). Concomitantly, upregulation of γH2AX, indicating accumulated DNA damage, and an increase in p16$^{Ink4a}$ were observed in senescent HSFs, MEFs, and aged brain tissues (*Figure 1a–c*). Knocking down *ATM* via shRNA accelerated senescence in HSFs, evidenced by increased β-galactosidase activity (*Figure 1d–e*), enlarged morphology (*Figure 1—figure supplement 2a*), accumulated γH2AX (*Figure 1f*), and reduced cell proliferation (*Figure 1—figure supplement 2b*). These data indicate that ATM decline retards DDR and drives senescence.

Other than DNA damage, ATM is activated by chloroquine (CQ), an antimalarial drug that modulates chromatin confirmation (*Bakkenist and Kastan, 2003*). We confirmed that a low dose of CQ increased the level of pS1981 auto-phosphorylation of ATM but not γH2AX (*Figure 1—figure supplement 2c*). We then investigated whether activating ATM by CQ can ameliorate senescence. As shown, the CQ treatment activated ATM (pS1981), promoted clearance of DNA damage (γH2AX), and inhibited apoptosis (cleaved Casp3) in HSFs (*Figure 1g*). Also, the CQ treatment suppressed β-galactosidase activity, which was abrogated if *ATM* was knocked down (*Figure 1h–i*). Importantly, CQ treatment extended the replicative lifespan of HSFs (*Figure 1j*). Likewise, CQ treatment activated Atm, cleared up accumulated DNA damage, suppressed β-galactosidase activity (*Figure 1k* and *Figure 1—figure supplement 2d–e*), and prolonged replicative lifespan in MEFs (*Figure 1l*). Although both 10 μM and 1 μM of CQ activated ATM, a dose-dependent toxicity assay showed that 1 μM is suitable for long-term treatment (*Figure 1—figure supplement 2f–g*). Of note, *ATM* KD or low dose of CQ applied in this study had little effect on basal autophagic activity (*Figure 1f,j* and *Figure 1—figure supplement 2g*). Collectively, CQ activates ATM to alleviate replicative senescence.

### An ATM-SIRT6 axis underlies age-associated metabolic reprogramming

A-T patients lacking functional ATM display features of premature aging, accompanied by insulin resistance and glucose intolerance (*Bar et al., 1978*; *Espach et al., 2015*). Senescent cells exhibit impaired mitochondrial respiration, but enhanced glycolysis producing more lactate (*Hagen et al., 1997*; *Lenaz et al., 2000*). As such, we wondered whether ATM decline triggers an age-associated metabolic shift. Levels of glycolytic genes *LDHB* and *PDK1* were dramatically increased in senescent MEFs and HSFs (*Figure 2a* and *Figure 2—figure supplement 1a*), and in liver tissues from *Atm-/-* mice (*Figure 2—figure supplement 1b*). Significantly, activating ATM via CQ suppressed senescence-associated glycolysis (*Figure 2a* and *Figure 2—figure supplement 1a*). Similarly, the

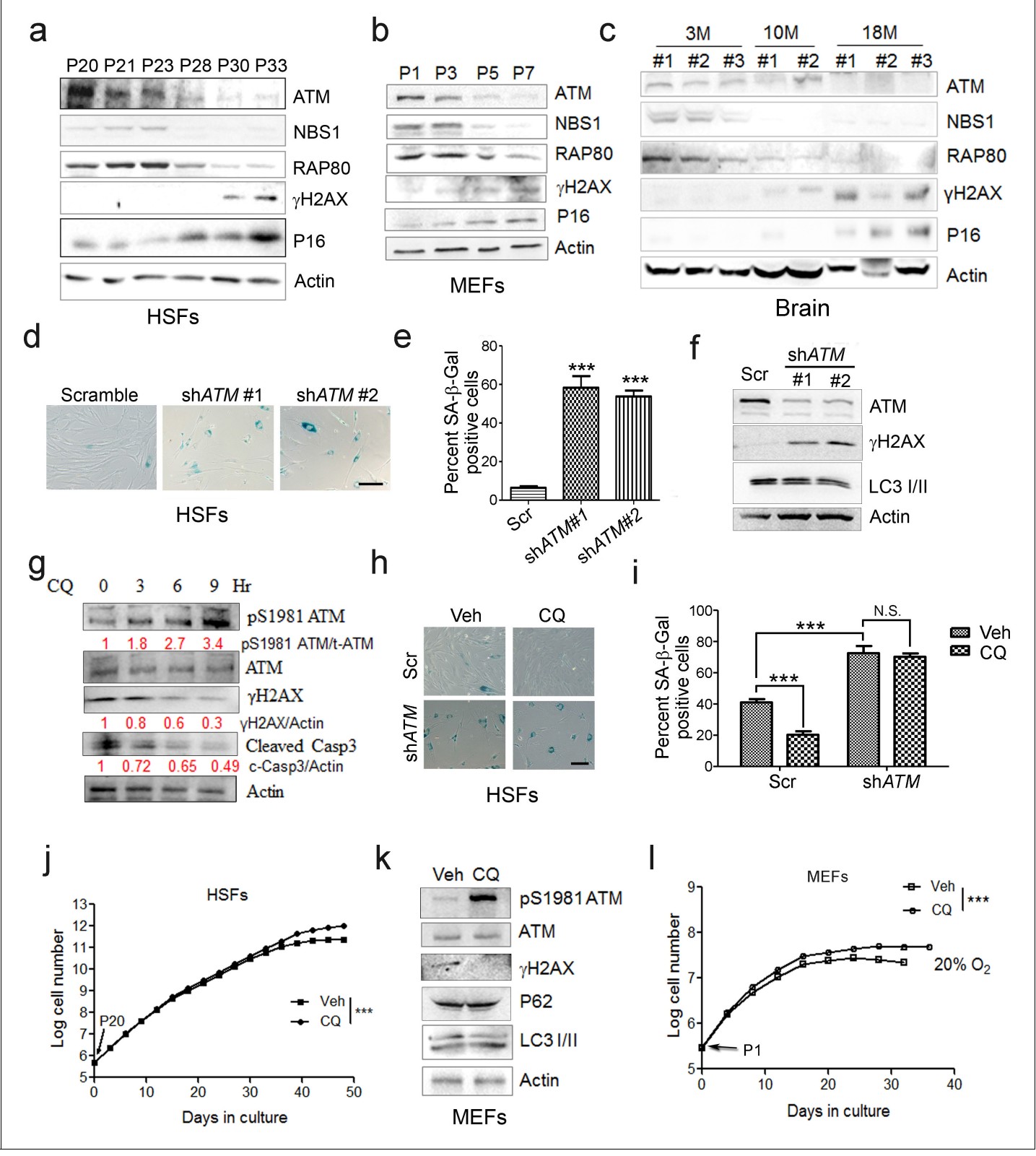

**Figure 1.** ATM activation by chloroquine alleviates senescence. (a) Immunoblots showing protein levels of ATM, NBS1, and RAP80 in human skin fibroblasts (HSFs). A gradually increased level of p16 indicates cellular senescence, while elevated γH2AX level indicates accumulated DNA damage. (b) Immunoblots showing protein levels of ATM, NBS1, and RAP80 in mouse embryonic fibroblasts (MEFs). (c) Immunoblots showing protein levels of ATM, NBS1, and RAP80 in brain tissues isolated from 3-, 10-, and 18-month-old male mice. (d) SA-β-Gal staining in HSFs treated with sh-*ATM* or scramble

*Figure 1 continued on next page*

*Figure 1 continued*

shRNA. Scale bar, 100 µm. (e) Quantification of SA-β-Gal-positive staining of (d) from five views randomly captured for each group. Data represent means ± SEM. ***p<0.001. (f) Immunoblots showing increased γH2AX and unaffected LC3I/II in HSFs treated with sh-*ATM* or scramble shRNA. (g) Immunoblots showing protein levels of pS1981 ATM, γH2AX, and cleaved caspase-3 in HSFs treated with 10 µM of CQ for indicated time. (h) SA-β-Gal staining in HSFs expressing either scramble or *ATM* shRNA treated with 1 µM CQ or DMSO (12 hr). Scale bar, 100 µm. (i) Quantification of SA-β-Gal-positive staining of (h) from five views randomly captured for each group. Data represent means ± SEM. ***p<0.001; 'N.S.' indicates no significant difference. (j) HSFs at passage 20 were continuously cultured with 1 µM CQ or DMSO, and cell number was calculated at each passage. Data represent means ± SEM. ***p<0.01. (k) Immunoblots showing protein levels of γH2AX, p62, and LC3 in MEFs treated with 1 µM CQ or DMSO. Note that CQ had little effect on the expression levels of p62 and LC3. (l) MEFs at passage one were continuously cultured in 20% $O_2$ with 1 µM CQ or DMSO, and cell number was determined at each passage. Data represent means ± SEM. ***p<0.01.

DOI: https://doi.org/10.7554/eLife.34836.003

The following source data and figure supplements are available for figure 1:

**Source data 1.** Statistical analysis for SA-β-Gal positive staining.

DOI: https://doi.org/10.7554/eLife.34836.006

**Source data 2.** Statistical analysis for EdU positive staining.

DOI: https://doi.org/10.7554/eLife.34836.007

**Figure supplement 1.** Decline of ATM-centered DNA repair machinery during senescence.

DOI: https://doi.org/10.7554/eLife.34836.004

**Figure supplement 2.** ATM regulates replicative senescence.

DOI: https://doi.org/10.7554/eLife.34836.005

inhibitory effect on glycolysis was diminished when *ATM* was depleted in HepG2 cells (*Figure 2b*). These data suggest a role for ATM in inhibiting glycolysis.

To examine how ATM regulates glycolysis, we performed RNA-Seq in *Atm-/-* MEF cells, and revealed a significant upregulation of glycolytic pathways (*Figure 2c*, and *Figure 2—source data 1*). Specific genes were validated by q-PCR (*Figure 2—figure supplement 1c*). As p53 is critical in glycolysis (*Kruiswijk et al., 2015*; *Schwartzenberg-Bar-Yoseph et al., 2004*), we further analyzed metabolomics of *Atm-/-* and control MEFs in *p53* null background. As shown, the metabolic profile exhibited a clear shift, i.e. mitochondrial electron transport chain and intermediates of TCA cycle were reduced, while intermediates of glycolysis were elevated (*Figure 2d*, *Figure 2—figure supplement 1d–e* and *Figure 2—source data 2*). The data suggest that *ATM* deficiency enhances anaerobic glycolysis in a p53-independent manner.

Sirt6 deacylase is able to shunt energy metabolism away from anaerobic glycolysis to the TCA cycle via H3K9ac-mediated local chromatin remodeling (*Sebastián et al., 2012*; *Zhong et al., 2010*). We noted that the level of H3K9ac was enhanced in cells depleted *ATM* (*Figure 2e*). Re-expressed ATM in A-T cells suppressed H3K9ac level (*Figure 2f*). ChIP analysis showed that H3K9ac was enriched at the promoter regions of glycolytic genes in *Atm-/-* cells (*Figure 2g*), where the relative occupancy of SIRT6 was abolished (*Figure 2h*). Consistent with increased H3K9ac, SIRT6 protein level was dramatically downregulated in *Atm-/-* mouse livers, and *ATM*-deficient HepG2, U2OS and HEK293 cells (*Figure 2—figure supplement 1f–i*). In contrast, protein levels of other sirtuins were not much affected in *ATM* KO HEK293 cells (*Figure 2i*), and mRNA levels of all sirtuins remained unchanged (*Figure 2—figure supplement 1j*). Moreover, transcriptomic analysis and q-PCR data illustrated that *Sirt6* depletion upregulated a similar cluster of genes essential for glycolysis (*Figure 2—figure supplement 2a–b* and *Figure 2—source data 3*). More importantly, the hyper-activated glycolytic pathway caused by *ATM* deficiency was completely restored by ectopic *SIRT6* in HepG2 cells (*Figure 2—figure supplement 2c*). The CQ treatment upregulated SIRT6 level and reduced H3K9ac level, especially at the regulatory regions of glycolytic genes (*Figure 2—figure supplement 2d–e*). Knocking down *SIRT6* abolished the inhibitory effect of CQ on glycolysis (*Figure 2b*). Additionally, *ATM* depletion in HEK293 cells, HSFs, and MEFs, significantly downregulated SIRT6 protein level, with little effect on SIRT1 or SIRT7 (*Figure 2i* and *Figure 2—figure supplement 2f–g*). Thus, these data suggest that ATM decline triggers an age-associated metabolic shift via SIRT6-mediated chromatin remodeling.

Other than metabolic abnormality, depleting *Sirt6* leads to premature aging features and shortened lifespan (*Mostoslavsky et al., 2006a*), whereas extra copies of *Sirt6* promote longevity in male mice (*Kanfi et al., 2012*). Given that Sirt6 was destabilized in *Atm* null mice, we wondered

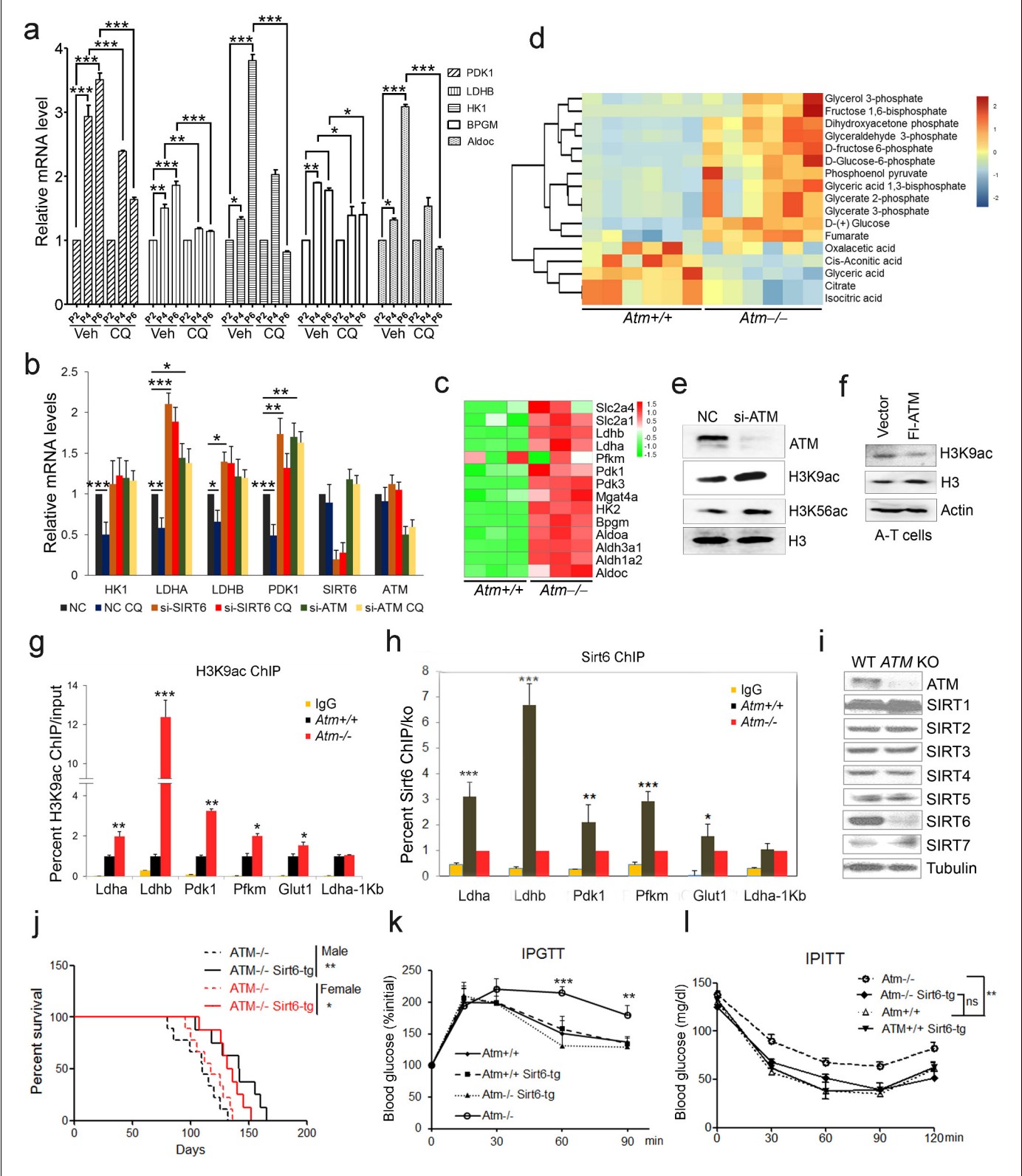

**Figure 2.** ATM-SIRT6 axis regulates age-related metabolic reprogramming. (a) Quantitative RT-PCR analysis of mRNA levels of indicated glycolytic genes in different passages of MEFs with or without treatment of CQ. Data represent means ± SEM. *p<0.05, **p<0.01, ***p<0.001. (b) Quantitative RT-PCR analysis of mRNA levels of indicated glycolytic genes in Scramble (NC), si-*SIRT6* or si-*ATM* HepG2 cells incubated with or without CQ (10 μM, 6 hr). Data represent means ± SEM. *p<0.05, **p<0.01, ***p<0.001. (c) Heatmap representation of RNA-Seq data (GSE109280) showing relative changes

*Figure 2 continued on next page*

*Figure 2 continued*

of glycolytic genes in *Atm-/-* MEF cells. The transcript levels are qualified in reads per kilobase of exon per million mapped sequence reads (RPKM), which is a normalized measure of exonic read density. Red and green indicate up- and downregulation, respectively. (**d**) Heatmap showing relative levels of metabolites in *Atm+/+* and *Atm-/-* MEF cells of p53 null background, analyzed by LC-MS. Red and blue indicate up- and downregulation, respectively. (**e**) Immunoblots showing protein levels of H3K9ac and H3K56ac in *ATM*-deficient HepG2 cells. (**f**) Immunoblots showing levels of H3K9ac in A-T cells reconstituted with Flag-ATM. (**g**) ChIP analysis showing enrichment of H3K9ac at the promoter regions of indicated genes in *Atm+/+* and *Atm-/-* MEFs. Data represent means ± SEM of three independent experiments. *p<0.05, **p<0.01, ***p<0.001. (**h**) ChIP analysis showing enrichment of Sirt6 at the promoter regions of indicated genes in *Atm+/+* and *Atm-/-* MEFs. Data represent means ± SEM of six independent experiments. *p<0.05, **p<0.01, ***p<0.001. (**i**) Immunoblots showing protein levels of sirtuins in wild-type (WT) and *ATM* knockout (KO) HEK293 cells. (**j**) Kaplan-Meier survival of *Atm-/-* and *Atm-/-;Sirt6*-Tg male (n = 11 in each group) and female (n = 9 in each group) mice. **p<0.01. (**k**) Results of glucose tolerance tests in *Atm+/+*, *Atm-/-*, and *Atm-/-;Sirt6*-Tg mice. Data represent means ± SEM, n = 6. **p<0.01, ***p<0.001. (**l**) Results of insulin tolerance tests in *Atm+/+*, *Atm-/-*, and *Atm-/-;Sirt6*-Tg mice. Data represent means ± SEM, n = 6. **p<0.01. 'ns' indicates no significant difference.

DOI: https://doi.org/10.7554/eLife.34836.008

The following source data and figure supplements are available for figure 2:

**Source data 1.** Differently expressed mRNA profiles of Atm-/- MEF cells.
DOI: https://doi.org/10.7554/eLife.34836.011
**Source data 2.** Differentially expressed Metabolites in Atm KO MEFs.
DOI: https://doi.org/10.7554/eLife.34836.012
**Source data 3.** Differently expressed mRNA profiles of Sirt6-/- MEF cells.
DOI: https://doi.org/10.7554/eLife.34836.013
**Figure supplement 1.** *Atm* deficiency promotes glycolysis.
DOI: https://doi.org/10.7554/eLife.34836.009
**Figure supplement 2.** SIRT6 reduction underlies age-related metabolic reprogramming triggered by ATM decline.
DOI: https://doi.org/10.7554/eLife.34836.010

whether the *Sirt6* transgene could rescue premature aging phenotypes and shortened lifespan in *Atm-/-* mice. To this end, we generated *Sirt6* transgenic mice by microinjection, and bred them with *Atm-/-* mice. The overexpression of Sirt6 was demonstrated by western blotting (***Figure 2—figure supplement 2h***). Significantly, ectopic *Sirt6* restored the elevation of serum lactate, and extended lifespan of *Atm-/-* mice of both genders (***Figure 2j*** and ***Figure 2—figure supplement 2i***). Importantly, *Atm-/-;Sirt6*-tg mice exhibited improved glucose tolerance and decreased insulin resistance (***Figure 2k–l***). Given that little difference was observed in glucose metabolism between young wild-type (WT) and *Sirt6*-transgenic mice (***Kanfi et al., 2012***), these data suggest a contributory role of the Atm-Sirt6 axis in the age-associated metabolic reprogramming.

## ATM phosphorylates and stabilizes SIRT6

Next, we examined how ATM regulates SIRT6. Significantly, overexpression of *ATM* increased SIRT6 level, but this was abolished when ATM was S1981A-mutated to block dimeric ATM dissociation (***Bakkenist and Kastan, 2003***; ***Berkovich et al., 2007***) (***Figure 3a***). Moreover, in addition to CQ, hypotonic buffer (20 mM NaCl), low glucose (LG), DNA-damaging agent camptothecin (CPT), and doxorubicin (Dox) all activated ATM and concomitantly increased SIRT6 protein level (***Figure 3—figure supplement 1a–c***), which was abrogated in *ATM*-depleted cells (***Figure 3—figure supplement 1b–c***). These data implicate a direct regulation of SIRT6 stability by ATM kinase activity. To confirm this, we first performed co-immunoprecipitation (Co-IP) in cells transfected with various FLAG-sirtuins. Interestingly, ATM was predominantly associated with SIRT6 among seven sirtuins (***Figure 3b***). The interaction was further confirmed at both ectopic and endogenous levels (***Figure 3c*** and ***Figure 3—figure supplement 1d***). Immunofluorescence microscopy showed co-localization of SIRT6 and ATM protein in the nucleus (***Figure 3d***). A domain mapping experiment indicated that the C-terminal domain was required for SIRT6 binding to ATM (***Figure 3—figure supplement 1e***). To determine whether ATM physically binds to SIRT6, 10 consecutive recombinant GST-ATM proteins were obtained and the binding to purified His-SIRT6 was analyzed. As shown (***Figure 3e***), His-SIRT6 bound predominantly to GST-ATM-4 (residues 770–1102) and relatively weakly to GST-ATM-1 (residues 1–250); both belong to the N-terminal HEAT repeat domain of ATM.

We next examined whether ATM phosphorylates SIRT6. Firstly, we found that CQ or CPT treatment significantly enhanced the binding of SIRT6 to ATM (***Figure 3f*** and ***Figure 3—figure***

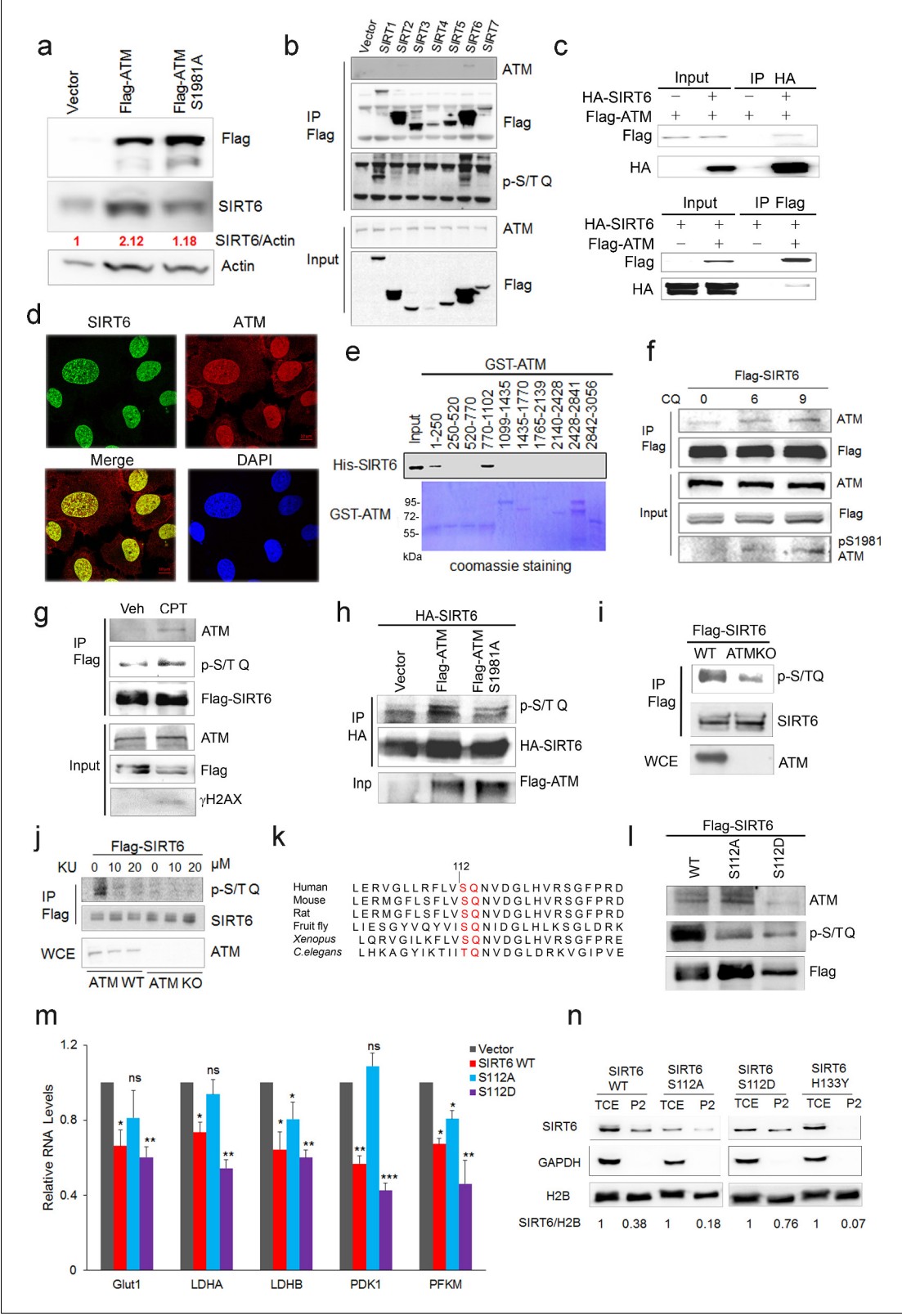

**Figure 3.** ATM interacts with and phosphorylates SIRT6. (**a**) Immunoblots showing protein levels of SIRT6 in HEK293 cells expressing Flag-ATM or Flag-ATM S1981A. (**b**) Immunoblots showing endogenous ATM and p-S/TQ motif in anti-Flag immunoprecipitates in HEK293 cells transfected with empty vector or Flag-sirtuins. (**c**) Immunoblots showing Flag-ATM and HA-SIRT6 in anti-HA (upper) or anti-Flag (lower) immunoprecipitates in HEK293 cells transfected with indicated constructs. (**d**) Representative photos of immunofluorescence staining of SIRT6 and ATM in U2OS cells, showing co-

*Figure 3 continued on next page*

Figure 3 continued

localization in the nucleus. Scale bar, 50 μm. (e) GST pull-down assay showing bacterially expressed His-SIRT6 predominantly bound to GST-ATM fragment 4 (770–1102), the N-terminal HEAT-repeat of ATM. (f) Immunoblots showing the increased binding capacity of ATM and SIRT6 under the treatment of (10 μM) CQ for the indicated time. (g) Immunoblots showing ATM and p-S/TQ in anti-Flag immunoprecipitates in HEK293 cells expressing Flag-SIRT6 treated with CPT (0.4 μM) or DMSO. (h) Immunoblots showing level of p-S/TQ SIRT6 in HEK293 cells co-transfected with HA-SIRT6 and Flag-ATM, Flag-ATM S1981A, or empty vector. (i) Immunoblots showing p-S/T Q SIRT6 in WT or *ATM* KO HEK293 cells. (j) Immunoblots showing p-S/T Q level of SIRT6 in *ATM* WT or KO HEK293 cells treated with DMSO and KU55933 (10 or 20 μM, 2 hr). (k) Alignment of protein sequence of human SIRT6 and orthologues in mouse, rat, fruit fly, *Xenopus*, and *C. elegans*. A conserved $S^{112}Q^{113}$ motif was highlighted. (l) Immunoblots showing p-S/T Q level of Flag-SIRT6, Flag-SIRT6 S112A, or Flag-SIRT6 S112D in HEK293 cells. (m) Quantitative RT-PCR analysis of mRNA levels of indicated glycolytic genes in sh-*SIRT6* HepG2 cells re-expressing SIRT6, SIRT6 S112A, or 112D mutation. Data represent means ± SEM. *p<0.05, **p<0.01, ***p<0.001. 'ns' indicates no significant difference. (n) Immunoblots showing SIRT6 protein level in total cell extract (TCE) and chromatin-enriched fractions (P2). Densitometry analysis was performed to determine the relative ratio of SIRT6/H2B within chromatin fractions.
DOI: https://doi.org/10.7554/eLife.34836.014

The following figure supplement is available for figure 3:

**Figure supplement 1.** ATM directly phosphorylates SIRT6.
DOI: https://doi.org/10.7554/eLife.34836.015

supplement 1f), whereas the S1981A mutant blocked such association (*Figure 3—figure supplement 1g*). ATM preferentially phosphorylates the S/T-Q motif. In the presence of CPT, an increased p-S/TQ level of SIRT6 was identified (*Figure 3g*). Of note, lambda protein phosphatase (λPP) diminished the p-S/TQ level of SIRT6 (*Figure 3—figure supplement 1h*). Likewise, the p-S/TQ level of SIRT6 was elevated in cells treated with low glucose, which activates ATM by ROS generation (*Assaily et al., 2011*; *Sarre et al., 2012*) (*Figure 3—figure supplement 1i*). Moreover, ectopic ATM significantly increased the p-S/TQ level of SIRT6, but this was abolished in the case of S1981A mutant (*Figure 3h*). Consistently, a pronounced reduction of p-S/TQ level of SIRT6 was observed in cells lacking *ATM* or treated with KU55933, a selective and specific ATM kinase inhibitor (*Berkovich et al., 2007*; *Hickson et al., 2004*) (*Figure 3i–j*). The decrease in p-S/TQ level was primarily attributable to loss of *ATM*, as it was restored by ectopic FLAG-ATM in a dose-dependent manner (*Figure 3—figure supplement 1j*). Indeed, SIRT6 has one evolutionarily conserved $S^{112}Q^{113}$ motif (*Figure 3k*). We therefore constructed S112A and S112D mutants, which resemble hypo- and hyper-phosphorylated SIRT6 respectively. As shown, these mutations almost abolished the pS/T-Q level of FLAG-SIRT6 (*Figure 3l*). The in vitro kinase assay showed that ATM could phosphorylate GST-SIRT6, but not S112A (*Figure 3—figure supplement 1k*). Furthermore, compared with SIRT6 S112A, ectopic S112D exhibited a much higher inhibitory effect on glycolytic gene expression in sh-*SIRT6* HepG2 cells (*Figure 3m* and *Figure 3—figure supplement 1l*), and enhanced chromatin association of SIRT6 (*Figure 3n*). Collectively, the data suggest that ATM directly phosphorylates SIRT6 at Serine 112.

We next examined whether ATM is involved in regulating SIRT6 protein stability. Notably, compared with WT or vehicle control, the degradation rate of ectopic and endogenous SIRT6 was largely increased in *ATM* KO HEK293 cells, *Atm-/-* MEFs, and cells incubated with KU55933 in the presence of cycloheximide (CHX) (*Figure 4a–b* and *Figure 4—figure supplement 1a–c*). Recently, MDM2 was demonstrated to ubiquitinate SIRT6 and promote its proteasomal degradation (*Thirumurthi et al., 2014*). We therefore examined the polyubiquitination level of SIRT6. As shown, the ubiquitination level of FLAG-SIRT6 in *ATM* KO cells was significantly elevated compared with WT (*Figure 4—figure supplement 1d*). While S112A mutant markedly enhanced the polyubiquitination level of SIRT6, S112D had little effect (*Figure 4—figure supplement 1e*). Moreover, S112A accelerated SIRT6 degradation, whereas S112D retarded it (*Figure 4c–d*), indicating that the Ser112 phosphorylation by ATM regulates SIRT6 ubiquitination and thus protein stability. Indeed, ectopic MDM2 enhanced the polyubiquitination level of FLAG-SIRT6 (*Figure 4—figure supplement 1f*). In the case of *ATM* depleted or SIRT6 S112A mutant, the binding capacity of SIRT6 to MDM2 was enhanced (*Figure 4e* and *Figure 4—figure supplement 1g*). In searching for key residues that are polyubiquitinated by MDM2, we identified two clusters of lysine residues, i.e. K143/145 and K346/349, which are conserved across species. We then generated KR mutations of these residues, and found that K346/349R remarkably reduced the polyubiquitination level of SIRT6 (*Figure 4—figure supplement 1h*). Individual KR mutation showed that K346R significantly blocked MDM2-mediated ubiquitination and

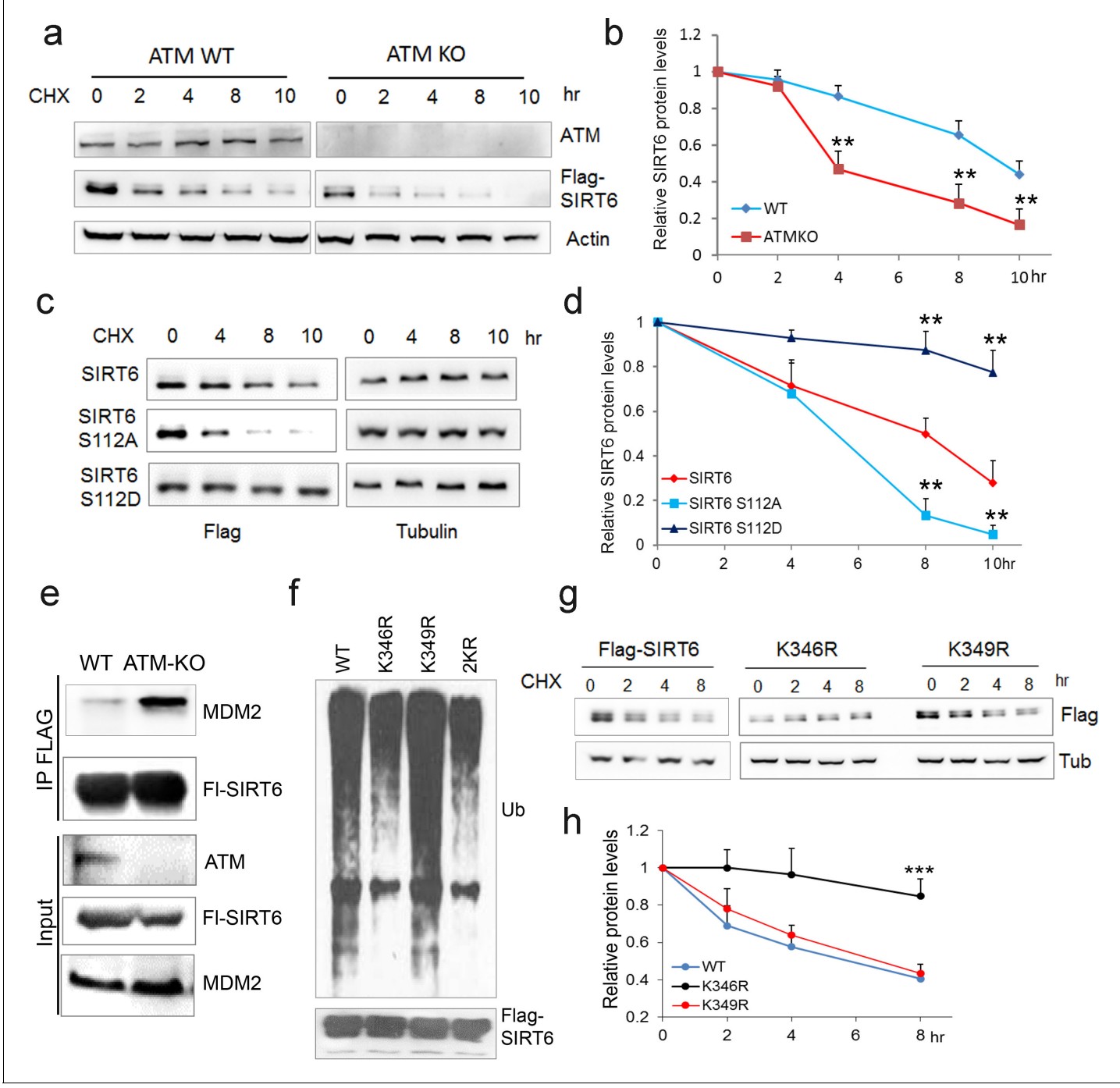

**Figure 4.** ATM prevents ubiquitination and degradation of SIRT6. (**a**) Immunoblots showing protein levels of Flag-SIRT6 in WT and *ATM* KO HEK293 cells treated with CHX (50 μg/ml) for indicated periods of time. (**b**) Quantification of protein levels in (**a**) by ImageJ. Data represent means ± SEM of three independent experiments. **p<0.01. (**c**) Immunoblots showing protein levels of Flag-SIRT6, S112A, and S112D in the presence of CHX (50 μg/ml) for indicated periods of time. (**d**) Quantification of protein levels in (**c**) by ImageJ. Data represent means ± SEM of three independent experiments. ***p<0.001. (**e**) Immunoblots showing increased binding capacity between SIRT6 and MDM2 in *ATM* KO HEK293 cells. (**f**) Immunoblots showing ubiquitination of Flag-SIRT6, K346R, K349R, and K346/349R (2KR) in HEK293 cells. Note that 2KR and K346R abrogated the ubiquitination of Flag-SIRT6. (**g, h**) Upper, immunoblots showing protein levels of Flag-SIRT6, K346R, and K349R in the presence of CHX (50 μg/ml) for indicated periods of time. Lower, quantification of protein levels by ImageJ. Data represent means ± SEM of three independent experiments. ***p<0.001.

DOI: https://doi.org/10.7554/eLife.34836.016

The following figure supplement is available for figure 4:

**Figure supplement 1.** ATM-mediated phosphorylation of SIRT6 prevents its ubiquitination and degradation.

*Figure 4 continued on next page*

*Figure 4 continued*

DOI: https://doi.org/10.7554/eLife.34836.017

degradation of SIRT6, whereas K349R hardly affected it (*Figure 4f–g*). More importantly, K346R restored the increased ubiquitination and accelerated protein degradation of SIRT6 S112A (*Figure 4—figure supplement 1i–j*). Collectively, these data indicate that K346 is subject to MDM2-mediated ubiquitination, which is inhibited by ATM-mediated S112 phosphorylation.

## Activating ATM via CQ promotes longevity

The cellular data suggest a pro-longevity function of ATM. We then tested it at organismal level. We employed *Caenorhabditis elegans,* which have a short lifespan of approximate 30 days. Nematodes deficient for *atm-1*, an orthologue of mammalian *ATM*, and WTs were exposed to various doses of CQ (see Materials and methods). Significantly, the period treatment with CQ (1.0 μM) extended the median lifespan (~14%) of *C. elegans* (*Figure 5a*). The lifespan-extending effect was abolished in *atm-1* KO (*Figure 5b*) or in SIRT6 homolog *sir-2.4* KD nematodes (*Figure 5—figure supplement 1a–b*). The data suggest that CQ promotes longevity in an ATM- and SIRT6- dependent manner. We further examined the beneficial effect of CQ in a HGPS model, i.e. *Zmpste24-/-* mice, which has a shortened lifespan of 4–6 months (*Pendás et al., 2002*) and impaired ATM-mediated DNA repair signaling (*Liu et al., 2013a*). We found that the level of Atm was dramatically reduced in *Zmpste24-/-* MEFs and tissues (*Figure 5c* and *Figure 5—figure supplement 1c*). Significantly, CQ treatment activated Atm, stabilized Sirt6, decreased the accumulated DNA damage, inhibited glycolysis, and alleviated senescence in *Zmpste24-/-* cells (*Figure 5d–e* and *Figure 5—figure supplement 1d–e*). The CQ treatment also delayed body weight decline, increased running endurance, and prolonged lifespan in *Zmpste24-/-* mice (*Figure 5f–h*), but had no significant effect on the lifespan of *Atm-/-* mice (*Figure 5—figure supplement 1f*).

Physiologically aged mice frequently develop aging-associated metabolic disorders, with high glucose and lactate (*Houtkooper et al., 2011*). Given that ATM declines with age, and activation of ATM by CQ inhibits glycolysis in senescent cells and *Zmpste24-/-* mice, we intraperitoneally administrated 12-month-old 'old' male mice with low-dose CQ (3.5 mg/kg) twice a week. Remarkably, compared with the saline-treated group, CQ treatment inhibited glycolysis, lowered serum lactate level, and attenuated body weight decline (*Figure 5i* and *Figure 5—figure supplement 1g–h*), implicating potential benefits of CQ in physiologically aged mice. Generally, these data demonstrate a lifespan-extending benefit of ATM activation by CQ.

## Discussion

DNA damage accumulates with age, and defective DDR and DNA repair accelerates aging. However, whether boosting DNA repair machinery promotes healthiness and longevity is still unknown. DNA damage stimulates DDR, but if persistent, it instead leads to senescence. Therefore, if enhancing DDR efficacy possibly promotes longevity, it must be DNA damage free. The antimalarial drug CQ can intercalate into the internucleosomal regions of chromatin, unwind DNA helical twist, and thus activate ATM without causing any DNA damage (*Bakkenist and Kastan, 2003*; *Krajewski, 1995*). We demonstrate that long-term treatment with CQ activates ATM, improves DNA repair, restores age-related metabolic shift, alleviates cellular senescence, and extends lifespan of nematodes and *Zmpste24* null mice. Mechanistically, ATM phosphorylates the longevity gene SIRT6 (*Tasselli et al., 2017*), and prevents MDM2-mediated ubiquitination and proteasomal degradation of SIRT6. To our knowledge, this is the first study to establish direct causal links between robust DNA repair machinery and longevity. In support of this notion, DNA repair efficacy has been shown to be enhanced in long-lived naked mole rat (*MacRae et al., 2015*), and human longevity has been shown to be associated with single nucleotide polymorphisms (SNPs) in DNA repair genes/pathways (*Debrabant et al., 2014*; *Soerensen et al., 2012*). Interestingly, the heterozygous rather than homozygous status of a SNP, albeit both enhance the transcription of *ATM*, is associated with longevity in Chinese and Italian populations (*Chen et al., 2010*; *Piaceri et al., 2013*). Therefore, in future study, it would be worthwhile evaluating whether *Atm* can promote longevity in model organisms, and, if so, how many extra copies are required.

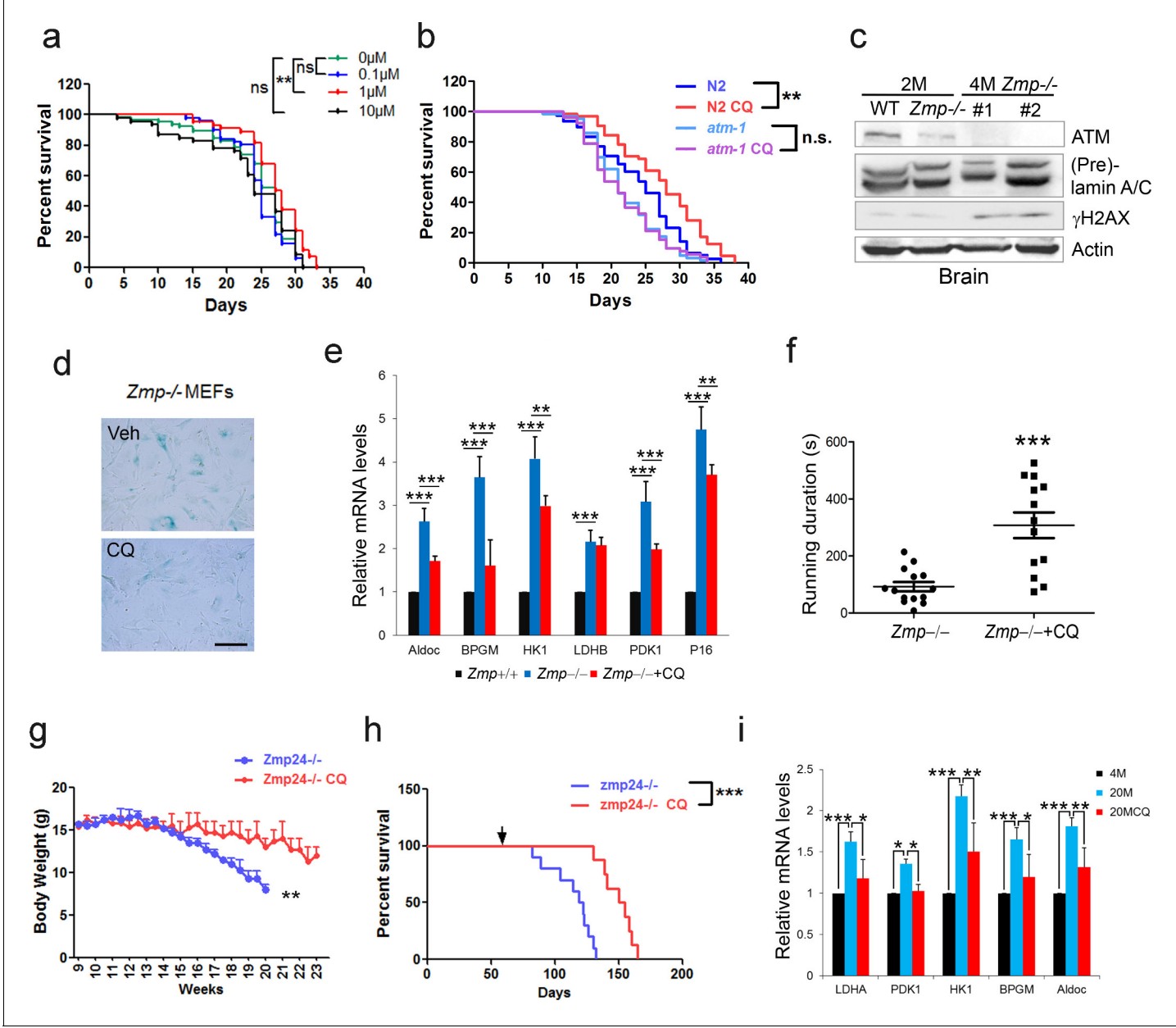

**Figure 5.** CQ extends lifespan in an ATM- dependent manner. (**a**) Survival analysis of *C. elegans* treated with the indicated dosage of CQ. **p<0.01. NS indicates no significant difference. (**b**) Survival analysis of wild-type and *atm-1* null *C. elegans* cultured in medium with or without 1 μM CQ. (**c**) Immunoblots showing protein levels of Atm and γH2AX in brain tissues of *Zmpste24+/+* (2 months), *Zmpste24-/-* (2 months), and *Zmpste24-/-* (4 months) mice. (**d**) Representative images showing SA-β-Gal staining in *Zmpste24-/-* MEFs with or without CQ treatment. Scale bar, 100 μm. (**e**) Quantitative RT-PCR analysis of mRNA levels of *p16^Ink4a* and indicated glycolytic genes in liver tissues of *Zmpste24+/+*, saline-treated, and CQ-treated *Zmpste24-/-* mice. Mice were treated for 8 weeks with two weekly intraperitoneal injections of CQ at 3.5 mg/kg. Data represent means ± SEM. *p<0.05, **p<0.01, ***p<0.001. (**f**) Maximum running duration in saline- and CQ-treated *Zmpste24-/-* mice. Data represent means ± SEM. ***p<0.001. (**g**) Body weight of saline- and CQ-treated male *Zmpste24-/-* mice. Data represent means ± SEM. **p<0.01. (**h**) Kaplan-Meier survival curves of saline-treated (n = 10) and CQ-treated (n = 8) *Zmpste24-/-* mice. ***p<0.001. (**i**) Quantitative RT-PCR analysis of mRNA levels of indicated glycolytic genes in the liver tissues of 4-month-old, saline-treated 12-month-old (n = 3), and CQ-treated 20-month-old (n = 3) mice. Data represent means ± SEM. *p<0.05, **p<0.01, ***p<0.001.

DOI: https://doi.org/10.7554/eLife.34836.018

The following source data and figure supplements are available for figure 5:

**Source data 1.** Lifespan analysis of chloroquine's effect on nematodes.

DOI: https://doi.org/10.7554/eLife.34836.021

*Figure 5 continued on next page*

*Figure 5 continued*

**Source data 2.** Lifespan analysis of chloroquine's effect on mice.
DOI: https://doi.org/10.7554/eLife.34836.022
**Figure supplement 1.** ATM activation ameliorates aging-associated features.
DOI: https://doi.org/10.7554/eLife.34836.019
**Figure supplement 2.** Schematic model of ATM-SIRT6 axis in regulating aging and longevity.
DOI: https://doi.org/10.7554/eLife.34836.020

Accumulation of DNA damage and metabolic disturbance are common denominators of aging (*López-Otín et al., 2013*; *Moskalev et al., 2013*). Metabolic reprogramming from the TCA cycle to glycolysis is prominent in both physiological and pathological aging (*Feng et al., 2016*; *Shimizu et al., 2014*). Why senescent cells become glycolytic is poorly understood. The crosstalk between cellular metabolism and DDR is not well elucidated. Upon genotoxic stress, ATM represses the rapamycin-sensitive mammalian target of the rapamycin (mTORC1) pathway (*Alexander et al., 2010*), but activates the pentose phosphate pathway (PPP) (*Cosentino et al., 2011*), suggesting that cell metabolism may be key downstream of DDR signaling. Moreover, it is recognized that deficiency in DNA repair machinery such as ATM, WRN, and Ercc1, accelerates aging and causes severe metabolic disorders (*Garinis et al., 2008*; *White and Vijg, 2016*). In this study, we showed that boosting ATM activity using a low dose of CQ enhances genomic stability, attenuates age-onset metabolic reprogramming, alleviates senescence, and extends lifespan in mice. The data demonstrate for the first time that enhanced DNA repair machinery (ATM-SIRT6 axis) promotes longevity. Considering that ATM and SIRT6 function not only in maintenance of genome integrity but also as homeostatic protein modifiers, the pro-longevity role of ATM is most likely benefited from enhanced DNA repair and metabolic homeostasis, but it is difficult to determine which is more important.

Recently, Bohr's group identified that increased consumption of $NAD^+$ by an early DDR factor poly (ADP-ribose) polymerase (PARP1), owing to accumulated DNA damage, accelerated aging in *Atm* mutant mice (*Fang et al., 2016*). $NAD^+$ serves as a cofactor of sirtuins, including SIRT1 and SIRT6. Therefore, this work establishes a linear causal link between deficient DDR, DNA damage accumulation, consumption of $NAD^+$, decline in sirtuin activity, and aging. Moreover, administration of nicotinamide mononucleotide or nicotinamide riboside ameliorates age-related function decline and extends lifespan in mice (*Mills et al., 2016*; *Zhang et al., 2016*). Here, we found that ATM decline during aging causes DNA damage accumulation and enhances glycolysis, both of which consume most of the $NAD^+$, providing an explanation for the low $NAD^+$ level in *Atm-/-* mice and physiologically aged mice.

Closely resembling normal aging, HGPS has attracted numerous efforts in understanding of molecular mechanisms and developing therapeutic strategies (*De Sandre-Giovannoli et al., 2003*; *Eriksson et al., 2003*). We and others have found that HGPS and *Zmpste24* null cells undergo premature senescence because of defective chromatin remodeling (*Ghosh et al., 2015*; *Krishnan et al., 2011*; *Liu et al., 2013a*; *Liu et al., 2013b*), delayed DDR, and impaired DNA repair (*Liu et al., 2005*; *Liu et al., 2013b*; *Varela et al., 2005*). Specifically, Atm-Kap-1 signaling is compromised (*Liu et al., 2013a*), and SIRT6 protein level and deacetylase activity are reduced in progeria cells (*Endisha et al., 2015*; *Ghosh et al., 2015*). Here we showed that Atm is significantly downregulated, which explains the reduced SIRT6, delayed DDR, and metabolic shift in progeria cells and mice. It would be interesting to investigate whether ectopic *Atm* or *Sirt6* could rescue progeroid features in these mice. Nevertheless, the activation of ATM via CQ remarkably improves glucose homeostasis, DNA damage clearance, and running endurance, and extends lifespan in progeria mice. It would be worthwhile evaluating the pro-longevity benefits of CQ in physiological aging.

CQ is an FDA-approved and clinically used medicine for treatment of malaria (2015). Via activation of ATM, long-term treatment of CQ protects against atherosclerosis, improves insulin sensitivity, and rescues glucose tolerance in type 2 diabetes (T2D) (*Emami et al., 1999*; *Razani et al., 2010*; *Schneider et al., 2006*). The lysosomotropic property of CQ also makes it a potent inhibitor of autophagy (*Yang et al., 2013*). The application of CQ for antimalarial treatment (500 mg/week, maximum 0.8 μM in plasma) and for cancer therapy (100–500 mg/day) (*Kimura et al., 2013*) is attributed to its inhibitory action on autophagy. Of note, CQ also the attenuates inflammatory response by inhibiting autophagy (*Szatmári-Tóth et al., 2016*; *Whelan et al., 2017*; *Wu et al., 2018*),

requiring a high dosage of 50 mg/kg for mice. In the current study, we used a low dose of CQ to activate ATM, i.e. 1–10 μM for cell line and 3.5 mg/kg twice a week for mice (*Schneider et al., 2006*). The results showed that at such low doses, CQ has no toxicity and little effect on basal autophagic activity. Moreover, a low dose of CQ prolongs lifespan in progeroid mice, but exhibits little effect on *Atm* KO background, supporting an ATM-dependent pro-longevity function of CQ. Unfortunately, we could not test the CQ effect in *Sirt6-/-* mice with only 1-month lifespan (*Mostoslavsky et al., 2006b*). Here, we addressed the pro-longevity benefits of CQ-activated ATM, most likely attributable to improved DNA repair and glucose metabolism. Given that ATM also displays anti-inflammatory function (*Erttmann et al., 2017*; *Shoelson, 2006*), we could not rule out an anti-inflammatory effect in lifespan extension observed in CQ-treated mice.

In conclusion, our data establish direct causal links between robust DNA repair machinery and longevity. In line with DNA damage theory of aging, we propose that DNA damage activates DDR; however, its constant activation causes senescence; defective ATM-SIRT6 axis underlies premature aging, exemplified by HGPS and A-T mouse models, which are rescued by treatment of CQ and *Sirt6* transgene, respectively; in physiological aging, DNA damage-free activation of ATM by CQ stabilizes SIRT6, thus promoting longevity in nematodes and most likely also in mice (*Figure 5—figure supplement 2*). Our findings provide a novel therapeutic strategy for HGPS, and could facilitate clinical trials of CQ as an effective treatment for age-related diseases.

# Materials and methods

### Key resources table

| Reagent type (species) or resource | Designation | Source or reference | Identifiers | Additional information |
|---|---|---|---|---|
| Gene (human) | ATM | National Center for Biotechnology Information | Gene ID: 472 | |
| Gene (mouse) | Atm | National Center for Biotechnology Information | Gene ID: 11920 | |
| Gene (*Caenorhabditis elegans*) | atm-1 | National Center for Biotechnology Information | Gene ID: 3565793 | |
| Gene (human) | SIRT6 | National Center for Biotechnology Information | Gene ID: 51548 | |
| Gene (mouse) | Sirt6 | National Center for Biotechnology Information | Gene ID: 50721 | |
| Gene (*Caenorhabditis elegans*) | sir-2.4 | National Center for Biotechnology Information | Gene ID: 182284 | |
| Gene (mouse) | Zmpste24 | National Center for Biotechnology Information | Gene ID: 230709 | |
| Gene (mouse) | p53 | National Center for Biotechnology Information | Gene ID: 230710 | |
| Cell line (human) | HEK293 | ATCC | Catalog number: ATCC CRL-1573; RRID:CVCL_0042 | |
| Cell line (human) | HepG2 | ATCC | Catalog number: ATCC HB-8065; RRID:CVCL_0027 | |
| Cell line (human) | U2OS | ATCC | Catalog number: ATCC HTB-96; RRID:CVCL_0042 | |
| Cell line (mouse) | *Atm-/-; p53-/-* MEF | from Dr. Yosef Shiloh (Tel Aviv University, Israel) | | |
| Cell line (mouse) | *Sirt6-/- MEF* | from Dr. Raul Mostoslavsky (Massachusetts General Hospital Cancer center, USA) | | |

*Continued on next page*

*Continued*

| Reagent type (species) or resource | Designation | Source or reference | Identifiers | Additional information |
|---|---|---|---|---|
| Antibody | ATM | Abcam (Cambridge, UK) | Cat# ab78; RRID:AB_306089 | Applications: WB; Immunofluorescence |
| Antibody | SIRT6 | Abcam (Cambridge, UK) | Cat# ab62739; RRID:AB_956300 | Applications: WB; Immunofluoresce; Chromatin immunoprecipitation |
| Antibody | γH2AX | Abcam (Cambridge, UK) | Cat# ab81299; RRID:AB_1640564 | Applications: WB |
| Antibody | p21 | Santa Cruz Biotechnology | Cat# sc-6246; RRID:AB_628073 | Applications: WB |
| Antibody | MDM2 | Santa Cruz Biotechnology | Cat# sc-965; RRID:AB_627920 | Applications: WB |
| Antibody | p-ATM (Ser1981) | EMD Millipore | Cat# 05–740; RRID:AB_309954 | Applications: WB |
| Antibody | H3K9ac | EMD Millipore | Cat# 07–352; RRID:AB_310544 | Applications: WB; Chromatin immunoprecipitation |
| Antibody | p-S/T Q | Cell Signaling Technology (Beverly, MA) | Cat #9607S; RRID:AB_10889739 | Applications: WB |
| Antibody | cleaved caspase-3 | Cell Signaling Technology (Beverly, MA) | Cat #9661; RRID:AB_2341188 | Applications: WB |
| Antibody | HA | Sigma-Aldrich | Cat# H3663; RRID:AB_262051 | Applications: WB |
| Antibody | Flag | Sigma-Aldrich | Cat# F1804; RRID: AB_262044 | Applications: WB |
| Antibody | LC3B | Sigma-Aldrich | Cat# L7543; RRID:AB_796155 | Applications: WB |
| Transfected construct (human) | Flag-His-ATM wt | Addgene (Cambridge, MA) | Cat #31985 | |
| Transfected construct (human) | Flag-SIRT6 | Addgene (Cambridge, MA) | Cat #13817 | |
| Transfected construct (human) | Flag-His-ATM S1981A | Addgene (Cambridge, MA) | Cat #32300 | |
| Commercial assay or kit | Senescence beta-galactosidase staining Kit | Cell Signaling Technology (Beverly, MA) | Cat #9860 | |
| Commercial assay or kit | Lactate Colorimetric Assay Kit | BioVision | Cat #K667-100 | |
| Commercial assay or kit | Click-iT EdU Alexa Fluor 488 Kit | Invitrogen | Cat #C10425 | |
| Chemical compound, drug | Cycloheximide | Sigma-Aldrich | Cat #66-81-9 | |
| Chemical compound, drug | MG-132 | Sigma-Aldrich | Cat #474787 | |
| Chemical compound, drug | Chloroquine | Sigma-Aldrich | Cat #C6628 | |

## Mice

*Zmpste24-/-* mice and *Atm-/-* mice have been described previously (*Barlow et al., 1996*; *Pendás et al., 2002*). *Sirt6*-transgenic mice (*Sirt6*-tg) of C57BL/6J background were constructed by injecting cloned mSirt6 cDNA with CAG promoter into fertilized eggs. Primers for genotyping of *Sirt6* transgenic allele were as follows: forward: 5'-CTGGTTATTGTGCTGTCTCATCAT-3'; reverse: 5'-CCGTCTACGTTCTGGCTGAC-3'. *Atm-/-* mice were crossed to *Sirt6*-tg mice to get *Atm-/-;Sirt6*-tg

mice. Chloroquine (CQ) experiments were conducted as described (*Schneider et al., 2006*). Briefly, 12-month-old wild-type C57BL/6J male mice, 2-month-old *Zmpste24-/-*, and *Atm-/-* male mice were administered with CQ (Sigma, St. Louis, MO) in 0.9% saline twice per week at 7 mg/kg body weight, and the control group was treated with saline alone. At least 8 weeks after treatment of CQ, mice were subjected to functional tests. Body weight and lifespan was recorded. The survival rate was analyzed using the Kaplan–Meier method and statistical comparison was performed using the Log-rank Test. Mice were housed and handled in the laboratory animal research center of Shenzhen University. All experiments were performed in accordance with the guidelines of the Institutional Animal Care and Use Committee (IACUC). The protocols were approved by the Animal Welfare and Research Ethics Committee of Shenzhen University (Approval ID: 201412023).

## *C. elegans* survival assay

*C. elegans* nematode survival assay was performed according to standard protocols (*Kenyon et al., 1993*). Briefly, wild-type and *atm-1* null nematodes (100 to 150 per group) synchronized to prefertile young adult stage were exposed to NGM plates containing the indicated dosage of CQ. After 1-day incubation, animals were transferred to fresh incubation plates without CQ for another 2 days. This procedure was repeated every 3 days. Nematodes that showed no response to gentle stimulation were recorded as dead. The survival data were analyzed using the Kaplan–Meier method and statistical comparison was performed using the Log-rank Test.

## Cell lines

HEK293 (CRL-1573), HepG2 (HB-8065), and U2OS (HTB-96) cells were purchased from ATCC. Human skin fibroblasts HSFs (F2-S) and primary MEFs were prepared as described previously (*Liu et al., 2005*). Immortalized *Atm-/-; p53-/-* and *Sirt6-/-* MEFs were provided as a kind gift from Dr. Yosef Shiloh (Tel Aviv University, Israel) and Dr. Raul Mostoslavsky (Massachusetts General Hospital Cancer center, USA), respectively. These cell lines were authenticated by short tandem repeat (STR) profile analysis and genotyping, and were mycoplasma free. Cells were cultured in Gibco DMEM (Life Technologies, USA) with 10% fetal bovine serum (FBS), 100 U/ml penicillin and streptomycin (P/S) at 37°C in 5% $CO_2$ and atmospheric oxygen conditions. For CQ experiments, cells were maintained in the medium containing 1 µM chloroquine for 12 hr, and then grown in new fresh medium for 48 hr.

## Plasmids

Human Flag-SIRT6, pcDNA3.1 Flag-ATM, Flag-ATM S1981A, and pcDNA3 human MDM2 were all purchased from Addgene (Cambridge, MA). Flag-SIRT6 with amino acid substitution mutations (S112A, S112D, K346R/K349R) were generated by PCR-based mutagenesis using pcDNA3-Flag-SIRT6 as a template and a QuikChange II site-directed mutagenesis kit (Agilent Technologies), following the manufacturer's instructions. Primer sequences for amino acid mutations of SIRT6 were as follows: SIRT6 S112A: (forward) 5'-cgtccacgttctgggcgaccaggaagcgga-3', (reverse) 5'-tccgcttcctggtcgcccagaacgtggacg-3'; SIRT6 S112D: (forward) 5'-ccgtccacgttctggtcgaccaggaagcggag-3', (reverse) 5'-ctccgcttcctggtcgaccagaacgtggacgg-3'; SIRT6 K346R: (forward) 5'-ggccttcacccttctgggggggtctgtg-3', (reverse) 5'-cacagacccccccagaagggtgaaggcc-3'; SIRT6 K349R: (forward) 5'-gccttggccctcacccttttggggggt-3', (reverse) 5'-acccccccaaaagggtgagggccaaggc-3. HA-tagged human SIRT6 plasmid was amplified from the respective cDNAs and constructed into pKH3-HA vector. To express four truncated forms of SIRT6 protein, HA-SIRT6 plasmid as a template was constructed by PCR-based deletion.

## Protein extraction and western blotting

For whole cell protein extraction, cells were suspended in five volumes of suspension buffer (20 mM Tris-HCl, pH 7.5, 150 mM NaCl, 1 mM EDTA, 1 mM DTT, protease inhibitor cocktail), and then five volumes of 2X SDS loading buffer were added and incubated at 98°C for 6 min. Mice tissues were homogenized with 1 ml of ice-cold tissue lysis buffer (25 mM TrisHCl, pH 7.5, 10 mM $Na_3VO_4$, 100 mM NaF, 50 mM $Na_4P_2O_7$, 5 mM EGTA, 5 mM EDTA, 0.5% SDS, 1% NP-40, protease inhibitor cocktail). After homogenization and sonication, lysates were centrifuged at 16,000 g for 15 min. The clean supernatant was carefully transferred to new tubes. Protein concentrations were determined

using a bicinchoninic acid (BCA) assay method (Pierce, Rockford, IL) and were normalized with lysis buffer for each sample. Samples were denatured in 1X SDS loading buffer by boiling at 98°C for 6 min. Proteins were separated by loading to SDS-polyacrylamide gels, and then were transferred to PVDF membrane (Millipore). The protein levels were determined by immunoblotting using respective antibodies. The ImageJ program was used for densitometric analysis of immunoblotting, and the quantification results were normalized to the loading control.

## Antibodies

Rabbit anti-SIRT6 (ab62739), ATM (ab78), SIRT1 (ab12193), γH2AX (ab81299), RAP80 (ab52893), Kap-1 (ab10484), and p-KAP-1 (Ser824, ab70369) antibodies were obtained from Abcam (Cambridge, UK). Anti-lamin A/C (sc-20681), p21 (sc-6246), MDM2 (sc-965), and P53 (sc-6243) antibodies were purchased from Santa Cruz Biotechnology. Rabbit anti-γH2AX (05–636), p-ATM (Ser1981) (05–740), histone H3 (07–690), anti-H3K56ac (07–677), and H3K9ac (07–352) antibodies were sourced from EMD Millipore. Mouse anti-p-ATM (Ser1981) (#5883), p-S/TQ (#9607), ubiquitin (#3936), and cleaved caspase-3 (#9661) antibodies were purchased from Cell Signaling Technology (Beverly, MA). Mouse anti- HA, Flag, rabbit anti-LC3B, and P62 antibodies were obtained from Sigma-Aldrich. Anti-Nbs1 (NB100-143) antibody was purchased from Novus Biologicals. Mouse anti-actin, tubulin antibodies were obtained from Beyotime. Anti-pS112 SIRT6 monoclonal antibodies were prepared by Abmart generated from a specific phosphorylated peptide (peptide sequence CLRFVS$_P$QNV).

## Protein degradation assay

HEK293 cells (WT and ATM-deficient cells) were transfected with Flag-SIRT6 alone or together with Mdm2. 48 hr later, the cells were treated with 50 μg/ml of cycloheximide (CHX, Sigma-Aldrich), a translation inhibitor. For endogenous SIRT6 protein degradation assay, *ATM* wild-type and null MEFs were grown in 6 cm plates, and were treated with 50 mg/ml CHX for indicated time points. Cells were collected and the protein levels were determined by western blotting, the subsequent quantification was performed with ImageJ software.

## In vivo ubiquitination assay

In vivo ubiquitination assay was performed by transfecting HEK293 cells in 6 cm dishes with 1 μg Myc-ubiquitin, 2 μg Flag-SIRT6 or its mutations, and/or 1 μg MDM2 vector. 48 hr after transfection, cells were lysed in the buffer (25 mM Tris-HCl pH 8.0, 250 mM NaCl, 10 mM Na$_3$VO$_4$, 1 mM EDTA, 10% glycerol, protease inhibitor cocktail, and 0.1 mM phenylmethylsulphonyl fluoride), and then incubated with Flag-M2 beads (Sigma-Aldrich) overnight at 4°C. Beads were washed with lysis buffer three times, bound proteins were eluted by adding 1.5 × SDS loading buffer. The ubiquitin levels were analyzed by immunoblotting.

## In vitro kinase assay

HEK293T cells were transfected with 10 μg of FLAG-ATM and then treated with CPT. Activated ATM was immune-purified from the cell extracts with FLAG beads (Sigma, M8823). GST-SIRT6 or the S112A mutant was purified from bacteria. Kinase reactions were initiated by incubating purified ATM with GST-SIRT6 in the kinase buffer with or without 1 mM ATP for 120 min at 30°C. After reaction, proteins were blocked by SDS loading buffer. The membrane was then subjected to western blotting with antibodies against p-S/TQ.

## Immunoprecipitation

Cells under indicated treatments were totally lysed in lysis buffer containing 20 mM HEPES, pH 7.5, 150 mM NaCl, 10 mM Na$_3$VO$_4$, 10% glycerol, 2 mM EDTA, protease inhibitor cocktail, and 0.1 mM phenylmethylsulphonyl fluoride. After sonication and centrifugation, the supernatant was collected and incubated with H3K9ac (Millipore, 2 μg/sample) overnight at 4°C with a gentle rotation. Protein A/G agarose (Pierce, 10 μl/sample) were added to the tubes and rotated at 4°C for 2 hr. Beads were precipitated by centrifugation at 1000 g for 15 s and washed three times with cold lysis buffer. The pellet was resuspended in 1.5 × SDS loading buffer and incubated at 98°C for 6 min. The supernatants were collected and used for western blotting.

## GST pull-down assay

A series of GST fusion proteins of truncated ATM, which together spanned the full length of ATM, were constructed into pGEX4T-3 vector. For GST pull-down, bacterially expressed 6 × His tagged SIRT6 was separately incubated with various GST-ATM fragments in a buffer of 150 mM NaCl, 20 mM Tris-HCl [pH 7.5], 5 mM MgCl$_2$, 0.2 mM EDTA, 10% glycerol, 0.2% NP-40, and protease inhibitors (Roche Complete). GST-fusion proteins were then precipitated by adding Glutathione Sepharose fast flow (GE Healthcare). After washing twice with TEN buffer (0.5% Nonidet P-40, 20 mM Tris-HCl [pH 7.4], 0.1 mM EDTA, and 300 mM NaCl), glutathione agarose beads were analyzed by western blotting and coomassie staining.

## RNA interference and shRNA lentiviral infection

Briefly, cells were transfected with small interfering RNAs (siRNAs) for 48 hr using Lipofectamine 3000 (Invitrogen, USA) according to the manufacturer's instructions. The siRNAs targeting human *ATM*, *SIRT6*, and *HDM2* were purchased (GenePharma, China) with sequences as follows, si-*ATM*#1: 5'-AA UGUCUUUGAGUAGUAUGUU-3' (*Zhou et al., 2003*); Si-*ATM*#2: 5'-AAGCACCAGUCCAGUA UUGGC-3' (*Zhang et al., 2005*); si-*SIRT6*#1: 5'-AAGAAUGUGCCAAGUGUAAGA-3'; si-*SIRT6*#2: 5'-CCGGCTCTGCACCGTGGCTAA-3'; si-*HDM2*#1: 5'-AACGCCACAAATCTGATAGTA-3'; si-*HDM2*#2: 5'-AATGCCTCAATTCACATAGAT-3'. A scrambled siRNA sequence was used as control. Lentiviral shRNA constructs were generated in a pGLVH1 backbone (GenePharma, China), and virus was produced in HEK293 cells. To deplete ATM in HSF cells and SIRT6 in HepG2 cells, lentiviral infection was performed in the presence of 5 μg/ml polybrene. Two days later, the infected HSF cells or HepG2 cells were selected with 2 μg/ml puromycin. To downregulate sir-2.4 expression, the NL2099 worms were exposed to incubation plates containing HT115 bacteria with sir-2.4 RNAi vector.

## CRISPR/Cas9-mediated genome editing

Gene mutagenesis by the CRISPR/Cas9 system was conducted as described (*Ran et al., 2013*). The following gRNAs targeting human *ATM*, *SIRT6* were constructed in pX459 vector (Addgene, #48139). sg*ATM* F: 5'-CACCGATATGTGTTACGATGCCTTA-3', R: 5'- AAACTAAGGCATCGTAA-CACATATC-3'. HEK293 cells were transfected with pX459 or pX459-gRNA using Lipofetamine 3000 Transfection Reagent according to the manufacturer's instructions. After 2-day culture, cells were selected with 2 μg/ml puromycin, six colonies were picked and grown to establish stable cell lines. The targeted mutations were identified by western blotting, and PCR-based sequencing.

## EdU (5-ethynyl-2'-deoxyuridine) incorporation assay

EdU incorporation assays were conducted in HSF cells to estimate cell proliferation using the Click-iT EdU Alexa Fluor 488 Kit (Invitrogen, USA). HSF cells, infected by the respective lentiviruses containing shNC and sh*ATM*, were cultured in a six-well plate containing the coverslips in the presence of 10 μM EdU for 12 hr. Cells were fixed in 3.7% formaldehyde followed by 0.5% Triton X-100 permeabilization, and then stained with Alexa Fluor picolyl azide. Five random views were captured to calculate the positive staining rate for each group.

## Growth curves and SA-β-gal assays

Cell population doublings were monitored using a Coulter Counter. SA-β-galactosidase assay in primary cells was performed using Senescence β-galactosidase staining Kit (#9860, CST) according to the manufacturer's instructions. Five views were captured randomly to calculate the positive staining rate for each group.

## RNA preparation and Real-Time qPCR

Total RNA was extracted from cells or mouse tissues using Trizol reagent RNAiso Plus (TaKaRa, Japan) following the phenol–chloroform extraction method. Purified total RNA was used to obtain cDNA using PrimeScript RT Master Mix (Takara, Japan) following this method: 37°C for 30 min, and 85°C for 5 s. The gene expression was analyzed with the CFX Connected Real-Time PCR Detection System (BioRad) with SYBR Ex Taq Premixes (Takara, Japan). Gene expression levels were normalized to actin.

## Glucose tolerance test

Mice were fasted overnight (6 p.m. to 9 a.m.), and D-glucose (2.5 g/kg body weight) was administrated intraperitoneally. Blood glucose levels were determined from tail vein blood using a glucometer (Onetouch ultravue, Johnson, USA) at 0, 30, 60, 90, and 120 min after D-glucose injection.

## Insulin tolerance test

Mice were fasted for 6 hr (8 a.m. to 2 p.m.), and recombinant human insulin (0.75 U/kg body weight) was administered intraperitoneally. Blood glucose levels were determined in tail vein blood using a glucometer (Onetouch ultravue, Johnson) at 0, 30, 60, 90, and 120 min after insulin injection.

## Lactate assay

Mouse serum was five-fold diluted, and lactate concentration was determined with the Lactate Colorimetric Assay Kit (BioVision).

## Endurance running test

*Zmpste24-/-* mice were treated for 8 weeks with chloroquine or saline before running on a Rota-Rod Treadmill (YLS-4C, Jinan Yiyan Scientific Research Company, Shandong, China) to test the effect of chloroquine on fatigue resistance. Mice were placed on the rotating lane, and the speed was gradually increased to 10 r/min. When mice were exhausted and safely dropped from the rotating lane, the time latency to fall was automatically recorded.

## Metabolite analysis

Wild-type and *ATM* KO cells were grown in normal medium for 24 hr, and methanol-fixed cell pellets were analyzed by a two liquid chromatography-tandem mass spectrometry (LC-MS) method as described (*Luo et al., 2007*).

## Immunofluorescence microscopy

The cells were fixed using 4% paraformaldehyde at room temperature for 15 min, permeabilized by 0.5% Triton X-100 at room temperature for 10 min, blocked using 10% FBS/PBS, and then incubated with primary antibodies diluted in PBS containing 2% BSA overnight at 4°C. The primary antibodies were detected using an Alexa-488-conjugated anti-mouse secondary antibody (Invitrogen). The nuclei were stained using DAPI in anti-fade mounting medium. Images were captured using a Zeiss LSM880 confocal/multiphoton microscope.

## ChIP assay

Cells were fixed in 1% formaldehyde for 10 min at room temperature. The crosslinking reaction was quenched with 0.125 M glycine. After washing with PBS, cells were lysed with lysis buffer (50 mM Tris·HCl pH 8.0, 2 mM EDTA, 15 mM NaCl, 1% SDS, 0.5% deoxycholate, protease inhibitor cocktail, 1 mM PMSF), followed by sonication and centrifugation. The supernatant was collected and precleared in dilution buffer (50 mM Tris-HCl pH 8.0, 2 mM EDTA, 150 mM NaCl, 1% Triton X-100) with protein A/G Sepharose and pre-treated salmon DNA. The precleared samples were incubated overnight with H3K9ac antibody (2 µg/sample, Millipore) or appropriate control IgGs (Santa Cruz), and protein A/G Sepharose (Invitrogen). After washing sequentially with a series of buffers, the beads were heated at 65°C to reverse the crosslink. DNA fragments were purified and analyzed. Real-time PCR was performed with primers as described (*Zhong et al., 2010*):

 LDHB-ChIP-5': AGAGAGAGCGCTTCGCATAG
 LDHB-ChIP-3': GGCTGGATGAGACAAAGAGC
 ALDOC-ChIP-5': AAGTGGGGCACTGTTAGGTG
 ALDOC-ChIP-3': GTTGGGGGATTAAGCCTGGTT
 PFKM-ChIP-5': TTAAGACAAAGCCTGGCACA
 PFKM-ChIP-3': CAACCACAGCAATTGACCAC
 LDHA-ChIP-5': AGGGGGTGTGTGAAAACAAG
 LDHA-ChIP-3': ATGGCTTGCCAGCTTACATC
 LDHA-ChIP-1Kb-5': TGCAAGACAAGTGTCCCTGT
 LDHA-ChIP-1Kb-3': GAGGGAATGAAGCTCACAGC

## Statistical analysis

Statistical analyses were conducted using two-tailed Student's *t*-test between two groups. All data are presented as mean ± S.D. or mean ± S.E.M. as indicated, and a *p* value < 0.05 was considered statistically significant.

## Acknowledgements

We thank Dr. Yosef Shiloh (Tel Aviv University, Israel) for *Atm-/-; p53-/-* MEF cells and Dr. Raul Mostoslavsky (Massachusetts General Hospital Cancer center, USA) for *Sirt6-/-* MEFs. This project is supported by research grants from the National Natural Science Foundation of China (81422016, 91439133, 81571374, 81501206, 81501210), National Key R and D Program of China (2017YFA0503900, 2016YFC0904600), Research Grant Council of Hong Kong (773313, HKU2/CRF/13G), Natural Science Foundation of Guangdong Province (2014A030308011, 2015A030308007, 2016A030310064) and Shenzhen Science and Technology Innovation Commission (CXZZ20140903103747568, JCYJ20160226191451487 and JCYJ20140418095735645) and Discipline Construction Funding of Shenzhen (2016).

## Additional information

### Funding

| Funder | Grant reference number | Author |
|---|---|---|
| Natural Science Foundation of Guangdong Province | 2016A030310064 | Minxian Qian |
| National Natural Science Foundation of China | 81501206 | Minxian Qian |
| National Natural Science Foundation of China | 81501210 | Mingyan Zhou |
| Natural Science Foundation of Guangdong Province | 2015A030308007 | Baoming Qin |
| Shenzhen Science and Technology Innovation Commission | JCYJ20140418095735645 | Zimei Wang |
| Research Grant Council of Hong Kong | HKU2/CRF/13G | Zhongjun Zhou |
| National Natural Science Foundation of China | 81422016 | Baohua Liu |
| Ministry of Science and Technology of the People's Republic of China | 2017YFA0503900 | Baohua Liu |
| Natural Science Foundation of Guangdong Province | 2014A030308011 | Baohua Liu |
| Shenzhen Science and Technology Innovation Commission | CXZZ20140903103747568 | Baohua Liu |
| National Natural Science Foundation of China | 91439133 | Baohua Liu |
| National Natural Science Foundation of China | 81571374 | Baohua Liu |
| Ministry of Science and Technology of the People's Republic of China | 2016YFC0904600 | Baohua Liu |
| Shenzhen Science and Technology Innovation Commission | JCYJ20160226191451487 | Baohua Liu |
| Research Grant Council of Hong Kong | 773313 | Baohua Liu |

| Discipline Construction Funding of Shenzhen | Baohua Liu |
|---|---|

The funders had no role in study design, data collection and interpretation, or the decision to submit the work for publication.

## Author contributions
Minxian Qian, Conceptualization, Data curation, Formal analysis, Investigation, Writing—review and editing; Zuojun Liu, Formal analysis, Investigation, Methodology; Linyuan Peng, Validation, Investigation; Xiaolong Tang, Baoming Qin, Guangming Wang, Methodology; Fanbiao Meng, Ying Ao, Mingyan Zhou, Ming Wang, Resources; Xinyue Cao, Project administration; Zimei Wang, Zhongjun Zhou, Jun Xu, Resources, Methodology, Writing—review and editing; Zhengliang Gao, Methodology, Writing—review and editing; Baohua Liu, Conceptualization, Resources, Formal analysis, Supervision, Funding acquisition, Writing—original draft, Project administration, Writing—review and editing

## Author ORCIDs
Minxian Qian (iD) http://orcid.org/0000-0002-1763-2325
Xiaolong Tang (iD) http://orcid.org/0000-0002-4744-5846
Zhongjun Zhou (iD) http://orcid.org/0000-0001-7092-8128
Jun Xu (iD) https://orcid.org/0000-0001-8565-1723
Baohua Liu (iD) http://orcid.org/0000-0002-1599-8059

## Ethics
Animal experimentation: Mice were housed and handled in the laboratory animal research center of Shenzhen University. All experiments were performed in accordance with the guidelines of the Institutional Animal Care and Use Committee (IACUC). The protocols were approved by the Animal Welfare and Research Ethics Committee of Shenzhen University (Approval ID: 201412023).

## Decision letter and Author response
Decision letter https://doi.org/10.7554/eLife.34836.029
Author response https://doi.org/10.7554/eLife.34836.030

# Additional files

## Supplementary files
• Transparent reporting form
DOI: https://doi.org/10.7554/eLife.34836.023

## Data availability
Sequencing data have been deposited in GEO under accession code GSE109280.

The following dataset was generated:

| Author(s) | Year | Dataset title | Dataset URL | Database, license, and accessibility information |
|---|---|---|---|---|
| Qian M, Liu B | 2018 | Boosting ATM activity alleviates ageing and extends lifespan in a mouse model of progeria. | https://www.ncbi.nlm.nih.gov/geo/query/acc.cgi?acc=GSE109280 | Publicly available at the NCBI Gene Expression Omnibus (accession no: GSE109280). |

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
