## [Decision Letter]

Thank you for submitting your article "Boosting ATM Activity Promotes Longevity in Nematodes and Mice" for consideration by *eLife*. Your article has been favorably evaluated by Jessica Tyler (Senior Editor) and three reviewers, one of whom, Matt Kaeberlein (Reviewer #1), is a member of our Board of Reviewing Editors.

The reviewers have discussed the reviews with one another and the Reviewing Editor has drafted this decision to help you prepare a revised submission.

Summary:

All of the reviewers recognized the important and novel insights provided in this manuscript linking ATM, SIRT6, and longevity, but that there are some key revisions that need to be made before publication. In particular, the effects reported from CQ treatment are quite interesting, but could be due to targets other than ATM. It would greatly strengthen the manuscript to show genetically that ATM activation/overexpression recapitulates at least some of the reported phenotypes. This may not be feasible in mice, but is doable in worms or cells.

Several additional controls are needed to demonstrate phosphorylation of SIRT6 by ATM, as noted by reviewer #2.

There is some question as to whether the increases in lifespan observed in the manuscript are due to improved DNA damage response or to metabolic changes. In the absence of additional data addressing these two possibilities, the authors are encouraged to interpret their data in a more agnostic way and consider both models. This should be done both in the text and in a revised model figure.

Use of different cell lines for different experiments is confusing and detracts from confidence in the results. For example, why were MEFs used for senescence but HSFs were used for other assays, and HepG2 for even other assays. Optimally, most or all of the experiments would be done in a single consistent cell line.

There are numerous errors in grammar and data presentation throughout the manuscript. Several of these are listed below, and the authors should take much more care in preparing the revised version of their manuscript. Careful proofreading for English sentence structure and phrasing is also important.

Essential revisions:

1) Title: The title is a bit misleading and implies that the study shows that boosting ATM activity promotes longevity in wild type mice. I don't think it's necessary for the authors to show lifespan extension in wild type mice for publication, but their title and Abstract need to be changed so that it is clear that the longevity effects in mice are limited to short-lived mouse models of disease.

2) The authors should explain why the replicative senescence experiments were done in MEFs rather than HSFs and perhaps repeat these experiments in HSFs. Did the experiment not work in HSFs? Showing that CQ can extend replicative lifespan in HSFs would seem to be more relevant than MEFs. I also worry that the replicative capacity of the MEFs here is so short. Was this done under atmospheric oxygen conditions and, if so, it needs to be repeated under lower oxygen.

3) I found the worm experiments to be the least convincing part of the manuscript. Drug studies in worms are complicated by potential effects on the bacterial food, which I don't think was addressed here. There is no biochemical evidence that CQ gets into worms or modifies activity of the worm ATM homolog. There is no evidence that the worm ATM homolog phosphorylates the worm SIRT6 homolog. Again, the story sticks together with the model, but there are a lot of apparent assumptions that need to be made here. I realize the tools aren't available to do some of this, but you could show that transgenic overexpression of ATM is sufficient to have effects similar to CQ and increase lifespan in a sir-2.4-dependent manner.

4) The part of the paper dealing with phosphorylation of SIRT6 by ATM is novel and interesting, but needs some additional controls.

a) Figure 3D shows that both SIRT6 and ATM are in the nucleus (not surprisingly), but it is hard to judge whether they are co-localized. Ether a higher resolution confocal microscope is needed, or the figure can be removed.

b) Figure 4C, D: the gel does not show any difference between SIRT6 WT and S112D, yet quantification on the right shows very large effect. The authors need to provide a gel that corresponds to the quantification.

c) It is important to knock-in the SIRT6 S112 mutations in a cell line and perform RNAseq (or qRT-PCR on glycolytic genes) to prove that the mutation indeed affects SIRT6 regulated genes.

5) In Figure 2—figure supplement 1C, Figure 2h and Figure 2—figure supplement 2B, there were not have enough biological replicates.

---

## [Author Response]

Essential revisions:1) Title: The title is a bit misleading and implies that the study shows that boosting ATM activity promotes longevity in wild type mice. I don't think it's necessary for the authors to show lifespan extension in wild type mice for publication, but their title and Abstract need to be changed so that it is clear that the longevity effects in mice are limited to short-lived mouse models of disease.

We appreciate the reviewers’ suggestion. We have now removed related data in the revised manuscript and corresponding figures as requested. We will investigate the effect of CQ on physiological ageing in an independent project. As suggested, we changed the title to “Boosting ATM activity alleviates ageing and extends lifespan in a mouse model of progeria”. Accordingly, the Abstract was re-written; the figure panels were reorganized (see revised Figure 5 and Figure 5—figure supplement 1).

2) The authors should explain why the replicative senescence experiments were done in MEFs rather than HSFs and perhaps repeat these experiments in HSFs. Did the experiment not work in HSFs? Showing that CQ can extend replicative lifespan in HSFs would seem to be more relevant than MEFs. I also worry that the replicative capacity of the MEFs here is so short. Was this done under atmospheric oxygen conditions and, if so, it needs to be repeated under lower oxygen.

We thank the reviewers for their comments and suggestions. We performed the requested experiments in HSFs and showed consistent result that CQ extends replicative lifespan (see revised Figure 1J). We did notice that MEFs have a very limited growth capacity and could be spontaneously immortalized at a low frequency as a result of prolonged passaging (Sun and Taneja, 2007). Indeed, to demonstrate that ATM declines with age and senescence, we utilized two cell lines, HSFs and MEFs (revised Figure 1A and 1B). We showed that activating ATM via CQ retards senescence in both HSFs and MEFs (revised Figure 1H, 1I and Figure 1—figure supplement 2D-E). Under atmospheric oxygen conditions, normal primary MEFs undergo 10 ± 2 population doublingsin vitro, and then rapidly enter a slow growth status that has been described as replicative senescence (Blasco et al., 1997; Rittling, 1996; Todaro and Green, 1963). As suggested, we examined the effect of CQ on senescence of MEFS at physiological (3%) oxygen level. Compared with 20% O_2_, MEFs in 3% O_2_ exhibit no remarkable signs of senescence even reaching passage 9 (see Author Response Image 1), which is indeed consistent with existing literature (Parrinello et al., 2003). As indicated, the cultures in 3% O_2_ can reach more than 60 population doublings without any senescence phenotypes. In current project, to address that CQ treatment alleviates senescence by activating ATM and promoting clearance of DNA damage, we think it is appropriate to examine the replicative lifespan of MEFs under standard culture conditions, which exhibit shortened lifespan predominantly triggered by DNA damage accumulation. Thus, the Author Response Image 1 was not included in the revised manuscript.

**Author response image 1. respfig1:** Physiological (3%) oxygen condition is inappropriate for MEFs culture to assess the replicative lifespan-extending effect of CQ. (**a**) MEFs at passage 1 were cultured in 20% O_2_ (black) or 3% O_2_ (grey) with treatment of 1 μM CQ (circles) or DMSO (squares), and cell number was determined at each passage. ****P* < 0.01. (**b**) Quantitative RT-PCR analysis of mRNA levels of *p16* and *p21* genes in different passages of MEFs with or without treatment of CQ in 20% O_2_ or 3% O_2_. ****P* < 0.001. ‘ns’ indicates no significant difference. (**c**) Representative images showing morphology of MEFs under the indicated culture conditions with or without CQ treatment. (**d**) Quantification of SA-β-Gal-positive cells under indicated culture conditions. Data represent the means ± SEM. ****P* < 0.001; ‘ns’ indicates no significant difference.

3) I found the worm experiments to be the least convincing part of the manuscript. Drug studies in worms are complicated by potential effects on the bacterial food, which I don't think was addressed here. There is no biochemical evidence that CQ gets into worms or modifies activity of the worm ATM homolog. There is no evidence that the worm ATM homolog phosphorylates the worm SIRT6 homolog. Again, the story sticks together with the model, but there are a lot of apparent assumptions that need to be made here. I realize the tools aren't available to do some of this, but you could show that transgenic overexpression of ATM is sufficient to have effects similar to CQ and increase lifespan in a sir-2.4-dependent manner.

We agree with the reviewers that drug studies in worms are complicated and there is lack of solid evidence to support that worm ATM could phosphorylate *sir2.4*. In this study, we performed CQ treatment on N2, *atm-1* KO and *sir2.4*-depleted nematodes, showing that CQ extends lifespan of N2 worms, but not that of *atm-1* KO or *sir2.4*-depleted worms. The parallel treatment largely precludes potential effects of CQ on the bacterial food. To further strengthen the conclusion, as the reviewers suggested, we attempted to investigate the lifespan-extending effect of extra copies of *ATM-1* in worms. Since the endogenous *AMT-1* gene promoter is unknown and a 2 kb region at the upstream of *ATM-1* ORF did not work in our test, we constructed a large plasmid pPD95_75 with GFP-tagged full-length *ATM-1* (~7kbp) under the promoter of *sir2.4* (Chiang et al., 2012). The results indeed showed significant effect on lifespan extension (see Author Response Image 2). Of note, the heterozygous instead of homozygous status of a SNP, albeit both enhance the transcription of ATM, is associated with longevity in Chinese and Italian populations (Chen et al., 2010; Piaceri et al., 2013). However, though GFP tag was detected, we were unable to determine the amount of ectopic *ATM-1* in transgenic worms. In future study, it would be worthwhile to evaluate how many extra copies of Atm could maximally promote longevity in model organisms.

For dosage-dependent studies, while both 10 μM and 1 μM of CQ activate ATM, only 1 μM of CQ was suitably applied in the studies, which promotes longevity in HSFs, *Zmpste24-/-* mice, and nematodes, but exhibits little effect in *ATM*-depleted mice and worms. Perhaps it is an important facet of the amount of ATM for pro-longevity, requiring further study. We discussed this in the revised manuscript.

*C. elegans* with conserved DDR mechanism seems to be as a promising model for the investigation of DNA repair (Jones et al., 2012; Lemmens and Tijsterman, 2011). *ATM-1* in *C. elegans*, the homolog of mammalian ATM, is known to function in DDR and DSB repair, and its activity and functional interactions are conserved between worms and humans (Lee et al., 2010; Lemmens and Tijsterman, 2011). Similar to A-T patients and *Atm-/-* mice, loss-of-function of ATM-1 results in genomic instability and shortened lifespan (Fang et al., 2016; Jones et al., 2012). To test the lifespan-extending effect of ATM at organismal level, we employed *Zmpste24-/-* mice with reduced ATM and DNA repair capacity*, ATM-/-* mice and wild-type nematodes for CQ administration. The results showed that CQ improves SIRT6 protein level and extends the lifespan of *Zmpste24-/-* mice and N2 worms, but not that of *Atm-/-* mice and *ATM-1* KO worms. Moreover, the cellular data indicated that CQ promotes replicative lifespan of HSFs in an ATM-dependent manner. Biochemical studies revealed that ATM phosphorylates SIRT6 at the conserved Ser112 residue, blocking MDM2-mediated proteasomal degradation. Generally, based on these studies, we rationally believe that our current data is adequate to support a pro-longevity role of ATM.

**Author response image 2. respfig2:** Lifespan analysis for worms overexpressing ATM-1. (**a**) The strategy for constructing ATM-1 overexpressing plasmid with GFP-tag. (**b**) Representative images showing the detectable GFP expression in worms. (**c**) Survival analysis of worm line expressing GFP-ATM-1. **P* < 0.05.

4) The part of the paper dealing with phosphorylation of SIRT6 by ATM is novel and interesting, but needs some additional controls.a) Figure 3D shows that both SIRT6 and ATM are in the nucleus (not surprisingly), but it is hard to judge whether they are co-localized. Ether a higher resolution confocal microscope is needed, or the figure can be removed.

We thank the reviewers for the suggestion. SIRT6 is a nuclear NAD^+^-dependent deacetylase; ATM kinase is highly enriched in nuclei and also detectable in cytoplasm (Gately et al., 1998), which are consistent with our immunostaining result using SIRT6 and ATM antibodies. We have now presented high resolution confocal images, clearly showing intracellular punctate co-localization of SIRT6 and ATM (revised Figure 3D).

b) Figure 4C, D: the gel does not show any difference between SIRT6 WT and S112D, yet quantification on the right shows very large effect. The authors need to provide a gel that corresponds to the quantification.

We are thankful to the reviewers for their careful reading. We now presented new Western blots in revised Figure 4C, showing more clearly that S112D significantly retards SIRT6 degradation.

c) It is important to knock-in the SIRT6 S112 mutations in a cell line and perform RNAseq (or qRT-PCR on glycolytic genes) to prove that the mutation indeed affects SIRT6 regulated genes.

We appreciate for the reviewers’ insightful comments. Deficiency of *ATM* or *SIRT6* enhances the expression of glycolytic genes, whereas re-expressed SIRT6 in *ATM*-depleted HepG2 rescues the increase of glycolysis. ATM directly interacts with SIRT6 and phosphorylates it at Ser112. It is important to check the inhibitory effect of SIRT6 S112 mutations on glycolytic gene expression. Unfortunately, we failed to knock-in the SIRT6 S112 mutations in HepG2 by CRISPR-Cas9 system. Instead, to address this issue, we have now provided additional evidence from a practical approach. We utilized *SIRT6*-depleted HepG2 cells (revised Figure 3—figure supplement 1L), generated by shRNA lentiviral system (see Materials and methods) and re-constituted with SIRT6 WT, S112A and S112D. As shown in revised Figure 3M, the qRT-PCR data demonstrated that SIRT6 S112D exhibits more inhibitory effect on glycolytic gene expression than S112A. In addition, sub-cellular fractionation experiment showed that S112D enhances the chromatin-binding capacity of SIRT6, whereas S112A abrogated it (see revised Figure 3N). Our data suggest that SIRT6 S112A loses the regulation on targeted genes due to reduced chromatin-binding affinity and protein stability. Additional data indicate that ATM depletion or S112A mutation promotes SIRT6 degradation via MDM2-mediated proteasomal pathway.

5) In Figure 2—figure supplement 1C, Figure 2h and Figure 2—figure supplement 2B, there were not have enough biological replicates.

According to the reviewer’s suggestion, we have done these experiments with more repeats (n = 6 for each group) and did statistical analysis as shown in revised Figure 2—figure supplement 1C, Figure 2H and Figure 2—figure supplement 2B. The details have been included in revised figure legends.